



# The SPARC water vapour assessment II:
# Profile-to-profile comparisons of stratospheric and lower mesospheric water vapour data sets obtained from satellites

Stefan Lossow[1], Farahnaz Khosrawi[1], Michael Kiefer[1], Kaley A. Walker[2], Jean-Loup Bertaux[3], Laurent Blanot[4], James M. Russell[5], Ellis E. Remsberg[6], John C. Gille[7,8], Takafumi Sugita[9], Christopher E. Sioris[10], Bianca M. Dinelli[11], Enzo Papandrea[11,12], Piera Raspollini[13], Maya García-Comas[14], Gabriele P. Stiller[1], Thomas von Clarmann[1], Anu Dudhia[15], William G. Read[16], Gerald E. Nedoluha[17], Robert P. Damadeo[6], Joseph M. Zawodny[6], Katja Weigel[18], Alexei Rozanov[18], Faiza Azam[18], Klaus Bramstedt[18], Stefan Noël[18], John P. Burrows[18], Hideo Sagawa[19], Yasuko Kasai[20], Joachim Urban[21†], Patrick Eriksson[21], Donal P. Murtagh[21], Mark E. Hervig[22], Charlotta Högberg[23], Dale F. Hurst[24], and Karen H. Rosenlof[24]

[1]Karlsruhe Institute of Technology, Institute of Meteorology and Climate Research, Hermann-von-Helmholtz-Platz 1, 76344 Leopoldshafen, Germany.
[2]University of Toronto, Department of Physics, 60 St. George Street, Toronto, ON M5S 1A7, Canada.
[3]LATMOS, CNRS/UVSQ/IPSL, Quartier des Garennes, 11 Boulevard d'Alembert, 78280 Guyancourt, France.
[4]ACRI-ST, 260 Route du Pin Montard, 06904 Sophia-Antipolis Cedex, France.
[5]Hampton University, Center for Atmospheric Sciences, 23 Tyler Street, Hampton, VA 23669, USA.
[6]NASA Langley Research Center, 21 Langley Boulevard, Hampton, VA 23681, USA.
[7]National Center for Atmospheric Research, Atmospheric Chemistry Observations & Modeling Laboratory, P.O. Box 3000, Boulder, CO 80307-3000, USA.
[8] University of Colorado, Atmospheric and Oceanic Sciences, Boulder, CO 80309-0311, USA.
[9]National Institute for Environmental Studies, Center for Global Environmental Research, 16-2 Onogawa, Tsukuba, Ibaraki 305-8506, Japan.
[10]Environment and Climate Change Canada, 4905 Dufferin Street, Toronto, ON M3H 5T4, Canada.
[11]Istituto di Scienze dell'Atmosfera e del Clima del Consiglio Nazionale delle Ricerche (ISAC-CNR), Via Gobetti, 101, 40129 Bologna, Italy.
[12]Serco SpA, Via Sciadonna, 24–26, 00044 Frascati, Italy.
[13]Istituto di Fisica Applicata del Consiglio Nazionale delle Ricerche (IFAC-CNR), Via Madonna del Piano, 10, 50019 Sesto Fiorentino, Italy.
[14]Instituto de Astrofísica de Andalucía (IAA-CSIC), Glorieta de la Astronomía, 18008 Granada, Spain.
[15]University of Oxford, Atmospheric Physics, Clarendon Laboratory, Parks Road, Oxford OX1 3PU, United Kingdom of Great Britain and Northern Ireland.
[16]Jet Propulsion Laboratory, 4800 Oak Grove Drive, Pasadena, CA 91109, USA.
[17]Naval Research Laboratory, Remote Sensing Division, 4555 Overlook Avenue Southwest, Washington, DC 20375, USA.
[18]University of Bremen, Institute of Environmental Physics, Otto-Hahn-Allee 1, 28334 Bremen, Germany.
[19]Kyoto Sangyo University, Faculty of Science, Motoyama, Kamigamo, Kita-ku, Kyoto 603-8555, Japan.
[20]National Institute of Information and Communications Technology (NICT), 20 THz Research Center, 4-2-1 Nukui-kita, Koganei, Tokyo 184-8795, Japan.
[21]Chalmers University of Technology, Department of Space, Earth and Environment, Hörsalsvägen 11, 41296 Göteborg, Sweden.
[22]GATS Inc., 65 South Main Street #5, Driggs, ID 83442, USA.
[23]Stockholm University, Department of Physical Geography, Svante-Arrhenius-väg 8, 10691 Stockholm, Sweden.



[24]NOAA Earth System Research Laboratory, Global Monitoring Division, 325 Broadway, Boulder, CO 80305, USA.
[†]deceased on 14 August 2014

**Correspondence:** Stefan Lossow (stefan.lossow@kit.edu)

**Abstract.** Within the framework of the second SPARC (Stratosphere-troposphere Processes And their Role in Climate) water vapour assessment (WAVAS-II), profile-to-profile comparisons of stratospheric and lower mesospheric water vapour were performed considering 33 data sets derived from satellite observations of 15 different instruments. These comparisons aimed to provide a picture of the typical biases and drifts in the observational database and to identify data set specific problems. The observational database typically exhibits the largest biases below $70\,\mathrm{hPa}$, both in absolute and relative terms. The smallest biases are often found between $50\,\mathrm{hPa}$ and $5\,\mathrm{hPa}$. Typically, they range from $0.25\,\mathrm{ppmv}$ to $0.5\,\mathrm{ppmv}$ (5% to 10%) in this altitude region, based on the 50% percentile over the different comparison results. Higher up, the biases are overall increasing with altitude but this general behaviour is accompanied by considerable variations. Characteristic values vary between $0.3\,\mathrm{ppmv}$ and $1\,\mathrm{ppmv}$ (4% to 20%). Obvious data set specific bias issues are found for a number of data sets. In our work we performed a drift analysis for data sets overlapping for a period of at least 36 months. This assessment shows a wide range of drifts among the different data sets that are statistically significant at the $2\sigma$ uncertainty level. In general, the smallest drifts are found in the altitude range between about $30\,\mathrm{hPa}$ to $10\,\mathrm{hPa}$. Histograms considering results from all altitudes indicate the largest occurrence for drifts between $0.05\,\mathrm{ppmv}\cdot\mathrm{decade}^{-1}$ and $0.3\,\mathrm{ppmv}\cdot\mathrm{decade}^{-1}$. Comparisons of our drift estimates to those derived from comparisons of zonal mean time series only exhibit statistically significant differences in slightly more than 3% of the comparisons. Hence, drift estimates from profile-to-profile and zonal mean time series comparisons are largely interchangeable. As for the biases, a number of data sets exhibit prominent drift issues. In our analyses we found that the large number of MIPAS data sets included in the assessment affects our general results as well as the bias summaries we provide for the individual data sets. This is because these data sets exhibit a relative similarity with respect to the remaining data sets, despite that the fact that they are based on different measurement modes and different processors implementing different retrieval choices. Because of that, we have by default considered an aggregation of the comparison results obtained from MIPAS data sets. Results without this aggregation are provided on multiple occasions to characterise the effects due to the numerous MIPAS data sets. Among other effects, they cause a reduction of the typical biases in the observational database.

# 1 Introduction

Water vapour in the stratosphere and lower mesosphere is important for a number of reasons. In the lower stratosphere water vapour is the most important greenhouse gas. As such it strongly affects global warming at the Earth's surface (Riese et al., 2012; Dessler et al., 2013). In addition, water vapour plays a decisive role for ozone chemistry (e.g. Solomon, 1999; Brasseur and Solomon, 2005). On one hand, water vapour is an essential component of polar stratospheric clouds (PSCs). The heterogenous chemistry occurring on the surfaces of the cloud particles causes the severe ozone depletion in the lower stratosphere





during winter- and springtime. On the other hand, water vapour is the primary source of hydrogen radicals (i.e. OH, H, $HO_2$). These radicals destroy ozone within auto-catalytic cycles and dominate the ozone budget in the lower stratosphere and above about $1\,\mathrm{hPa}$. Beyond that, water vapour is a particularly suitable trace gas to diagnose dynamical processes in the stratosphere such as the Brewer-Dobson circulation and the overturning circulation in the mesosphere (e.g. Brewer, 1949; Remsberg et al.,

1984; Mote et al., 1996; Pumphrey and Harwood, 1997; Seele and Hartogh, 1999).

In the stratosphere and lower mesosphere water vapour has two major sources. One is the transport of water vapour from the troposphere into the stratosphere, for which several pathways exist (Holton et al., 1995; Moyer et al., 1996; Fueglistaler et al., 2009; Sioris et al., 2016). The primary pathway is the slow ascent through the cold tropical tropopause layer, typically accompanied by large horizontal motions. The cold point temperature along the air parcel trajectories controls the amount

of water vapour entering the stratosphere. Another pathway is the convective lofting of ice. Once the ice particles reach the stratosphere they evaporate and correspondingly increase the amount of water vapour. A third pathway is the transport along isentropic surfaces that span both the troposphere and stratosphere. Occasionally water vapour is directly injected into the stratosphere by volcanic eruptions. Overall, about $3.5\,\mathrm{ppmv}$ to $4.0\,\mathrm{ppmv}$ of water vapour enter the stratosphere (Kley et al., 2000). The other major source is the in situ oxidation of methane. The importance of this process for the water vapour budget

increases with altitude and typically maximises in the upper stratosphere (Le Texier et al., 1988; Frank et al., 2018). In the lower mesosphere the methane abundances are small, so that its oxidation can no longer contribute significantly to the water vapour production. Above that, the oxidation of molecular hydrogen is a minor source of water vapour in the upper stratosphere and lower mesosphere (Sonnemann et al., 2005; Wrotny et al., 2010). The major sink of water vapour in the stratosphere is the reaction with $O(^1D)$. With increasing altitude photodissociation becomes increasingly important as a sink and plays the

dominant role in the mesosphere. Dehydration, the permanent removal of water due to the sedimentation of PSC particles in the polar vortices, is another sink, however its importance is limited in space and time (Kelly et al., 1989; Fahey et al., 1990). Leaving this last sink process aside, the volume mixing ratio of water vapour generally increases with altitude in the stratosphere due to the dominant role of methane oxidation. Usually, around the stratopause a maximum in the vertical distribution is found. Higher up, the volume mixing ratio of water vapour typically decreases since a major source is missing.

Satellite observations of water vapour in the stratosphere and lower mesosphere were performed since the second half of the1970s, with a few gaps. First sensible results could be derived from observations of the LIMS (Limb Infrared Monitor of the Stratosphere; Remsberg et al., 1984) and SAMS (Stratospheric and Mesospheric Sounder; Munro and Rodgers, 1994) instruments. Both were deployed on the Nimbus-7 satellite that was launched in October 1978. The LIMS observations of stratospheric water vapour lasted until May 1979 while the SAMS observations yielded results in the upper half of the strato-

sphere and lower mesosphere from 1979 to 1981. In the 1980s observations of the SAGE II (Stratospheric Aerosol and Gas Experiment II; Rind et al., 1993; Taha et al., 2004) and the ATMOS (Atmospheric Trace Molecule Spectroscopy; Gunson et al., 1990) instruments provided stratospheric water vapour information. The SAGE II instrument was carried by the Earth Radiation Budget Satellite (ERBS) and operated for almost 21 years from October 1984 to August 2005. In contrast, the first ATMOS observations covered only a short period of time from late April to early May in 1985. The instrument was part

of the European Space Agency's (ESA) Spacelab 3 laboratory module carried by the Space Shuttle. In September 1991 the





Upper Atmosphere Research Satellite (UARS) was launched. It carried four instruments that measured water vapour in the stratosphere and lower mesosphere, i.e. CLAES (Cryogenic Limb Array Etalon Spectrometer; Roche et al., 1993), HALOE (Halogen Occultation Experiment, Harries et al., 1996 or Kley et al., 2000), ISAMS (Improved Stratospheric and Mesospheric Sounder; Goss-Custard et al., 1996) and MLS (Microwave Limb Sounder; Lahoz et al., 1994). The HALOE observations lasted

until November 2005, providing many new insights on stratospheric and mesospheric water vapour. The observations of the other instruments were much more short-lived. The CLAES and ISAMS observations ceased May 1993 and July 1992, respectively. The MLS instrument operated longer, however the water vapour channel already ceased functioning in April 1993. In March/April 1992, April 1993 and November 1994 the ATMOS instrument performed more measurements, again aboard the Space Shuttle. During all these three missions, the MAS (Millimeter-wave Atmospheric Sounder; Bevilacqua et al., 1996)

instrument also obtained information on stratospheric and lower mesospheric water vapour. In addition, on the last of these three Space Shuttle flights water vapour observations by the CRISTA (Cryogenic Infrared Spectrometers and Telescopes for the Atmosphere; Offermann et al., 2002) and the MARSHI (Middle Atmosphere High Resolution Spectrograph Investigation; Summers et al., 2001) instruments were also carried out. In August 1997 CRISTA and MARSHI were put on a second Space Shuttle mission. From October 1996 to June 1997 the Improved Limb Atmospheric Sounder (ILAS; Kanzawa et al., 2002)

aboard the Advanced Earth Observing Satellite (ADEOS) performed observations of stratospheric water vapour at high latitudes. Similar coverage was obtained by the POAM III (Polar Ozone and Aerosol Measurement III; Nedoluha et al., 2002) instrument that was carried by the French SPOT 4 (Satellite Pour l'Observation de la Terre). The satellite was launched in March 1998 and POAM III delivered data until December 2005.

In 2000, within the framework of the first SPARC water vapour assessment (Kley et al., 2000), many of these satellite data

sets (i.e. LIMS, SAGE II, ATMOS, HALOE, MLS, MAS, ILAS, POAM III) were evaluated. The comparisons indicated a reasonable degree of consistency among the data sets in the stratosphere. On average, the majority of them showed biases of less than $\pm$ 10% (see Sect. 2.4, Fig. 2.72 and Tabs. 2.5 to 2.7 of Kley et al., 2000) relative to the HALOE data set, which was used as reference. The differences were typically larger in the altitude range between $100\,\mathrm{hPa}$ and $60\,\mathrm{hPa}$ than in the stratosphere higher up.

Since this first assessment a wealth of new satellite data sets focusing on stratospheric and lower mesospheric water vapour has been obtained. In 2001 the Odin, TIMED (Thermosphere-Ionosphere-Mesosphere Energetics and Dynamics) and Meteor-3M satellites were launched. Aboard they carried the SMR (Sub-Millimetre Radiometer; Urban et al., 2007), the SABER (Sounding of the Atmosphere using Broadband Emission Radiometry; Feofilov et al., 2009) and the SAGE III (Thomason et al., 2010) instruments, respectively. While the SMR and SABER instruments still perform observations of stratospheric

and mesospheric water vapour to this day, the SAGE III observations in the stratosphere ceased like those of POAM III in December 2005. In March 2002 Envisat (Environmental Satellite) was launched carrying three instruments performing water vapour observations in the stratosphere and lower mesosphere, namely GOMOS (Global Ozone Monitoring by Occultation of Stars; Montoux et al., 2009), MIPAS (Michelson Interferometer for Passive Atmospheric Sounding; Payne et al., 2007; Wetzel et al., 2013 and von Clarmann et al., 2009) and SCIAMACHY (Scanning Imaging Absorption Spectrometer for Atmospheric

Chartography, Noël et al., 2010; Azam et al., 2012; Weigel et al., 2016). The observation of all three instruments ceased in April





2012 when contact with the satellite was lost. Aboard ADEOS-II the successor of ILAS, i.e. ILAS II (Griesfeller et al., 2008), was also sent into orbit in 2002. As for ILAS, the observations were short-lived, effectively covering the time period from April to October 2003. The same year the Canadian SCISAT (or SCISAT-1) was launched. The satellite carries the ACE-FTS (Atmospheric Chemistry Experiment - Fourier Transform Spectrometer; Nassar et al., 2005) and MAESTRO (Measurement of

Aerosol Extinction in the Stratosphere and Troposphere Retrieved by Occultation; Sioris et al., 2010) instruments that perform observations to the present day. The ACE-FTS observations yield water vapour information in the stratosphere and mesosphere while those by MAESTRO cover the lower stratosphere. To this day also a new version of the MLS instrument performs observations of water vapour in the stratosphere and mesosphere (Waters et al., 2006). The instrument is deployed on the Aura satellite that was launched in July 2004. Aboard Aura there is a second instrument that was capable of observing water vapour

in the lower stratosphere, i.e. HIRDLS (High Resolution Dynamics Limb Sounder; Gille et al., 2013). Its operations ceased in March 2008 after an instrumental failure. Since April 2007 the SOFIE (Solar Occultation for Ice Experiment; Rong et al., 2010) instrument carried by the AIM (Aeronomy of Ice in the Mesosphere) satellite performs observations focusing on high latitudes. The penultimate addition to the observational database regarding lower stratospheric water vapour came from the SMILES (Superconducting SubMillimeter-Wave Limb-Emission Sounder; Baron et al., 2011) instrument that was mounted on

the International Space Station (ISS) in 2009. The observations by this instrument lasted until April 2010. Finally, in February 2017 an almost exact replica of the SAGE III instrument flown on the Meteor-3M satellite was carried to the ISS from where this new instrument performs observations of stratospheric water vapour.

Many of the satellite water vapour data sets obtained since the new millennium have been validated individually in the last years. Prominent examples can be found in the works of Carleer et al. (2008); Milz et al. (2009); Noël et al. (2010); Rong et al.

(2010); Sioris et al. (2010); Thomason et al. (2010); Azam et al. (2012) and Weigel et al. (2016). Within the framework of the second SPARC water vapour assessment (WAVAS-II) satellite observations of stratospheric and lower mesospheric water vapour obtained between 2000 and 2014 are collectively evaluated with respect to a multitude of parameters, like biases, drifts or variability characteristics (Lossow et al., 2017; Nedoluha et al., 2017; Khosrawi et al., 2018). This aims to gain a contemporary overview of the typical uncertainties in the observational database. As part of this programme we present here

profile-to-profile comparisons of more than 30 satellite data sets of stratospheric and lower mesospheric water vapour. The advantage of this approach is that it reduces the sampling error relative to comparisons of binned data sets, e.g. zonal or monthly means as used in the works of Hegglin et al. (2013); Lossow et al. (2017) and Khosrawi et al. (2018). Unlike the first SPARC water vapour assessment, we do not invoke a specific reference data set (which was HALOE) but compare all possible combinations of data sets. Besides biases we also focus on drifts among the data sets. The aim of this work is two-fold. On

one hand we want to provide a general overview of the typical biases and drifts in the observational database. On the other hand we also want to give an account of data set specific characteristics that could be valuable in the analysis of individual data sets. The outline of this work is as follows. In the next section we provide a very brief overview of the data sets considered and their handling. The comparison approach is described in detail in Sect. 3. The results are presented in Sects. 4 and 5. The former section focuses on biases and the latter section on drifts between the different data sets. Conclusions from this work are

provided in Sect. 6. Additional results are presented in the Supplement, complementing those of the main manuscript.



## 2 Data sets

In the present comparisons overall 33 data sets from 15 individual satellite instruments are considered. Table 1 lists them alphabetically with respect to the instrument name. In case of multiple data sets from one instrument, the data sets have been sorted either alphabetically (e.g. MIPAS-Bologna data sets before MIPAS-ESA data sets) or chronologically (e.g. ACE-

FTS v2.2 before ACE-FTS v3.5) or a combination of both. The table also lists the corresponding data set labels and numbers that are used in the figures. In addition, Fig. 1 provides a visual overview of the temporal coverage of the individual data sets to give an indication when coincident observations between two data sets were possible. A complete description of the individual data sets is provided in the WAVAS-II data set overview paper by Walker and Stiller (in preparation). The focus of the present comparisons is on observations that were acquired since the previous millennium as a follow-up to the last WAVAS report in

2000 (Kley et al., 2000). HALOE, POAM III and SAGE II have provided data in the old millennium but correspondingly those were not considered here. While the SABER observations cover almost the entire time period considered in the assessment no data set has become available and thus they are not part of WAVAS-II. Also, the SAGE III observations from the ISS are not considered as they only commenced in 2017.

    In a first step we screened the data sets according to the recommendations provided by the individual data set teams. Those

screening criteria are listed in full detail in the WAVAS-II data set overview paper (Walker and Stiller, in preparation). In addition, we excluded profiles from the comparison that exhibited volume mixing ratios below $-20\,\mathrm{ppmv}$ or above $50\,\mathrm{ppmv}$ anywhere at altitudes above $70\,\mathrm{hPa}$. This wide interval was chosen to reject obvious outliers that might influence the comparisons in an undesirable way and that were not removed by the earlier screening. For many data sets this affected only a handful profiles. In absolute numbers most profiles were affected for the GOMOS, HIRDLS, MIPAS-Bologna V5R NOM, MIPAS-

Oxford and SMR 544 GHz data sets. For the GOMOS data set this meant that about 3.5% of the profiles were discarded, for the other data sets the percentage was in the per mille range. As a last step we sorted the individual observations of a given data set chronologically.

## 3 Approach

### 3.1 Determination of coincident observations

Principally, we have considered observations from two data sets as coincident when the following criteria were satisfied:

- a maximum temporal separation of $24\,\mathrm{h}$

- a maximum spatial separation of $1000\,\mathrm{km}$

- a maximum latitude separation of $5°$

- a maximum equivalent latitude separation of $5°$




When different versions of the ACE-FTS, MIPAS, SCIAMACHY solar occultation and the SMILES data sets were compared with each other these coincidence criteria were not invoked. In these cases the exact same observations were compared. For SMR the different data sets are obtained on different measurement days, so that this exception does not apply. The same is true for MIPAS observations in the nominal mode (NOM) and the middle atmosphere (MA) mode. Also, the different

SCIAMACHY observation geometries did not provide simultaneous measurements among them.

To apply the equivalent latitude criterion a scalar value has been assigned to every observation. This value was based on an average of equivalent latitudes within the altitude range from 425 K to 2000 K potential temperature, which essentially covers the entire stratosphere. The equivalent latitude information was derived from MERRA (Modern Era Retrospective-Analysis for Research and Applications, Rienecker et al., 2011) reanalysis data of potential vorticity.

To determine the coincidences we went through the individual observations of the first data set and determined the observations of the second data set that fulfilled the coincidence criteria. If multiple coincidences were found we chose the coincidence closest in spatial distance. This choice is optimised for the stratosphere where the diurnal variation is small (Haefele et al., 2008). Close to the tropopause and towards the middle mesosphere the diurnal variation in water vapour becomes more relevant. Once an observation of the second data set was determined as a coincidence it was not considered any further as a possible

coincidence for other observations of the first data set. Inherent in this approach is that the final coincidence pairs are dependent on the choice of the first data set, i.e. comparing ACE-FTS vs. HALOE for example can result in different coincidences than comparing HALOE vs. ACE-FTS. To avoid inconsistent results based on this aspect, we derived only coincidences for the lower half of the data set comparison matrix and used those results for the upper half of that matrix. According to the sorting of the data sets in Tab. 1 the ACE-FTS v2.2 data set has been used as first data set in all comparisons. The SMR 489 GHz data

set was considered as first data set only in the comparison to the SOFIE data set while the latter never served as the first data set. We investigated the influence of the first data set choice based on test comparisons to the HALOE, ACE-FTS v2.2 and MIPAS-IMKIAA V5H data sets. Typically the differences in the biases were smaller than 0.05 ppmv or 1% in absolute and relative terms, respectively. Larger deviations were mostly found at the lower boundaries of the comparisons.

### 3.2   Consideration of different vertical resolutions

The data sets considered in our comparisons have different vertical resolutions. A summary figure and a description how the resolutions have been estimated is provided in the data set overview paper by Walker and Stiller (in preparation). Differences in the vertical resolution only play a role for the comparisons at altitudes where the vertical distribution exhibits distinct structures, elsewhere the data sets can be compared directly regardless of the resolution differences. In our work this concerns first and foremost the hygropause region in the lowermost stratosphere. To decide in which comparisons a consideration of differences

in the vertical resolution is necessary we categorised the data sets into various classes according to their vertical resolution $dz$ around the hygropause, using some reasonably selected resolution intervals. These classes are given by the first four columns in Tab. 2. The lower the class number the better the vertical resolution of the data sets around the hygropause. The differences in the vertical resolution were considered in those comparisons where the two data sets were in different classes. The data set in the lower class was degraded to the vertical resolution of the data set in the higher class. In the table columns some data sets





have been marked by an asterisk, indicating that these data sets have a limited coverage of the hygropause. Hence, comparisons to these data sets may not need the consideration of differences in the vertical resolution in this altitude range. Yet, they have been taken into account for completeness.

The water vapour maximum in the vicinity of the stratopause is relatively broad and can be accordingly considered less
problematic. Yet, some data sets exhibit a strong degradation of their vertical resolution in this altitude region, in particular in the lower mesosphere. To check any influence of this degradation we considered a fifth convolution class that includes data sets whose vertical resolution exceeds $6\,\mathrm{km}$ anywhere above $1\,\mathrm{hPa}$ in the resolution summary figure presented by Walker and Stiller (in preparation). The differences in the vertical resolution are considered in the comparisons to those data sets that are not part of this convolution class and which cover altitudes up to at least $1\,\mathrm{hPa}$. The GOMOS, HIRDLS, MAESTRO,
SCIAMACHY limb, SMILES-NICT band A, SMILES-NICT band B and SMR $544\,\mathrm{GHz}$ data sets do not fulfil the latter criterion.

Due to the focus on differences in the vertical resolution in two different altitude regions, hybrid cases are possible, i.e. comparisons between data sets where one data set is better vertically resolved around the hygropause but worse than the other data set at high altitudes and vice versa. In total there have been 19 such cases in which we performed two comparisons
considering individually the differences around the hygropause and at high altitudes. The results will be presented later as a combination of these two comparisons. Up to $10\,\mathrm{hPa}$ data from the comparison considering the differences in the vertical resolution around the hygropause are taken into account, above the results from the comparison focusing on the resolution differences at the stratopause and the lower mesosphere are used.

The degradation of the higher vertically resolved data sets followed the approach by Connor et al. (1994). Using the averaging
kernel $\mathbf{A}$ and the a priori profile $\boldsymbol{x}_{\mathrm{a\,priori}}$ of the lower resolved profile, which we denote collectively as convolution data, the degradation of the higher resolved profile $\boldsymbol{x}_{\mathrm{high}}$ can be achieved as follows:

$$\boldsymbol{x}_{\mathrm{deg}} = \boldsymbol{x}_{\mathrm{a\,priori}} + \mathbf{A} \cdot (\boldsymbol{x}_{\mathrm{high}} - \boldsymbol{x}_{\mathrm{a\,priori}}). \tag{1}$$

The degraded profile $\boldsymbol{x}_{\mathrm{deg}}$ can then be compared directly to the lower vertically resolved data set. For some data sets the averaging kernel considers the log space, i.e. $\mathbf{A} = \mathbf{A}_{\mathrm{ln}}$, based on a different retrieval approach. In these cases Eq. 1 has to be
adapted to (e.g. Stiller et al., 2012a):

$$\boldsymbol{x}_{\mathrm{deg}} = \exp\left\{\ln(\boldsymbol{x}_{\mathrm{a\,priori}}) + \mathbf{A}_{\mathrm{ln}} \cdot [\ln(\boldsymbol{x}_{\mathrm{high}}) - \ln(\boldsymbol{x}_{\mathrm{a\,priori}})]\right\}. \tag{2}$$

The third column of Tab. 3, which lists the sources and characteristics of the convolution data employed in our comparisons, indicates the data sets for which this aspect had to be considered. Please note, that this is specific to the convolution data employed here. For example, the retrievals of the MIPAS-Oxford V5H and V5R MA data sets are performed in log space. But
for these data sets we had to generate the convolution data ourselves (as described below, see the second column of Tab. 3) which simply assumed a linear space. The degradation of the vertically higher resolved data sets has been performed in the



natural domain of the lower resolved data sets, as specified in the fourth column of Tab. 3. Most data sets have volume mixing ratio as natural domain, only some SCIAMACHY data sets use number density. Again this is specific to the convolution data used in this work. The retrievals of the GOMOS and SCIAMACHY solar Onion data sets for example use number density as natural domain, but once more we needed to generate the corresponding convolution data which assumed volume mixing ratio

as the natural domain. Temperature and pressure data for the conversion between volume mixing ratio and number density have been provided by all data set teams, either retrieved from the same set of measurements or from an auxiliary data source as reanalysis. Walker and Stiller (in preparation) provide a comprehensive summary on the retrieval spaces and domains of the individual data sets as well as the sources of the additional temperature and pressure information.

Another aspect is, that the convolution data often exceed the altitude range covered by the particular profile to be degraded.

This can be handled by either reducing the altitude range of the convolution data or by extending the altitude range of the profile to be degraded. For that a priori or other climatological data as well as model simulations can be employed. In practice the latter approach is often the better choice, leading to more reasonable results at the vertical boundaries of the degraded profile. After the degradation the extension data is removed again. Here we utilised offset-corrected, climatological data from HAMMONIA (Hamburg Model of the Neutral and Ionized Atmosphere, Schmidt et al., 2006) as function of month and latitude.

The second column of Tab. 3 lists the sources of the convolution data that have been employed in the comparisons. For most MIPAS data sets and the SCIAMACHY limb data set the complete set of averaging kernels and the corresponding a priori data were available. A single characteristic averaging kernel and observation-dependent a priori data were provided for the MLS, SCIAMACHY lunar and solar OEM data sets. For the MIPAS-Oxford V5R NOM data set and both SMR data sets collections of characteristic averaging kernels were supplied. They are dependent on time and latitude band. For the SMR 544

GHz data set there is also a dependency on the tropopause altitude. This data set only covers the upper troposphere and lower stratosphere and the tropopause altitude is the main source of kernel variability. Since for the SMR data sets the convolution data are not saved by default we re-retrieved the convolution data from at least 20 (50) observations that fell into the individual bins (monthly and $20°$latitude, see Tab. 3) for the 544 GHz (489 GHz data) set. For those bins where overall fewer observations exist we re-retrieved all of them. From this set we selected as most representative convolution data the one where the averaging

kernel minimised the following equation:

$$\sum_{j=l_{\mathrm{start}}}^{l_{\mathrm{end}}} \left[ \boldsymbol{A}_d(j) - \overline{\boldsymbol{A}_d}(j) \right]^2 \qquad (3)$$

Here $\boldsymbol{A}_d$ denotes the averaging kernel diagonal that has been interpolated on a regular altitude grid prior to the analysis. $\overline{\boldsymbol{A}_d}$ is the average averaging kernel diagonal over the entire set of re-retrieved data for a particular bin and $j$ is the index over the altitude levels $l_{\mathrm{start}}, ..., l_{\mathrm{end}}$ that were considered. For the 544 GHz data set we took into account the altitude range between

10 km and 25 km, while for the 489 GHz data set the altitude range between 15 km and 50 km was considered.

For the remaining data sets averaging kernels are typically not part of their retrieval or could not be provided as for the MIPAS-Oxford V5H and V5R MA data sets. In these cases we generated averaging kernels ourselves based on Gaussian





functions, using volume mixing ratio as natural domain (as noted above) and kept the a priori constant at zero. The averaging kernel row $\boldsymbol{A}_r(j)$ for a given altitude index $j$ was calculated as follows:

$$\boldsymbol{A}_r(j) = \frac{\boldsymbol{G}(j)}{\sum_{j=1}^{n_a} \boldsymbol{G}(j)}, \tag{4}$$

with

$$\boldsymbol{G}(j) = \exp\left\{ -\frac{4 \cdot \ln(2) \cdot [\boldsymbol{z} - \boldsymbol{z}(j)]^2}{\boldsymbol{dz}(j)^2} \right\}. \tag{5}$$

In the equation $n_a$ represents the number of altitudes contained in the altitude vector $\boldsymbol{z}$. Accordingly $\boldsymbol{z}(j)$ is the altitude for which the averaging kernel row is calculated and $\boldsymbol{dz}(j)$ describes the vertical resolution at this altitude. The vertical resolutions that have been used to generate the averaging kernels of the individual data sets are also given in the second column of Tab. 3. For the MIPAS-Oxford V5H and V5R MA data sets the vertical resolutions have been assumed while for the other data sets

they are typically based on the field of view. The only exceptions are the GOMOS and the SCIAMACHY solar Onion data sets. For the latter the vertical resolution is based on the smoothing of the absorption profiles while the estimate for the GOMOS data set relied on actual averaging kernels. As altitude vector we considered the altitudes given in the data files for the individual observations. For the ACE-FTS data sets we used the data files with the tangent altitude grid and not those with the interpolated regular 1 km grid. When generating the averaging kernel for a given observation we set rows to zero for those altitudes where

data were missing, either due to lacking coverage or screening.

### 3.3   Derivation of biases between the data sets

The comparisons followed essentially the approach outlined by Dupuy et al. (2009), which compared various ozone data sets. The bias $\bar{b}(t, \phi, z)$ between two coincident data sets for a given bin of time $t$ and latitude $\phi$ and for a specific altitude $z$ has been calculated as

$$\bar{b}(t, \phi, z) = \frac{1}{n_c(t, \phi, z)} \cdot \sum_{i=1}^{n_c(t, \phi, z)} b_i(t, \phi, z), \tag{6}$$

where $n_c(t, \phi, z)$ denotes the corresponding number of coincident measurements and $b_i(t, \phi, z)$ are the individual differences between those. These differences were considered both in absolute

$$b_i(t, \phi, z) = b_{i,\mathrm{abs}}(t, \phi, z) = x_i(t, \phi, z)_1 - x_i(t, \phi, z)_2 \tag{7}$$

and relative terms

$$b_i(t, \phi, z) = b_{i,\mathrm{rel}}(t, \phi, z) = \frac{x_i(t, \phi, z)_1 - x_i(t, \phi, z)_2}{[x_i(t, \phi, z)_1 + x_i(t, \phi, z)_2]/2}, \tag{8}$$





where $x_i(t,\phi,z)_1$ are the individual water vapour abundances of the first data set and $x_i(t,\phi,z)_2$ correspondingly the abundances of the second data set. As denominator for the relative bias we used the mean of the two data sets. One common argument for this approach has been convenience as satellite observations can have larger uncertainties (Randall et al., 2003). Here, we also wanted to avoid intentionally any preference towards a certain data set as reference (as done in the first SPARC

water vapour assessment, see Introduction) to compare all data sets on equal terms.

Before the mean bias $\bar{b}(t,\phi,z)$ was derived we performed an additional screening on the individual biases $b_i(t,\phi,z)$ using the median and median absolute deviation (MAD, e.g. Jones et al., 2012). After screening profiles with data points outside a reasonable abundance range, as described in Sect. 2, this is a second attempt to ensure meaningful bias estimates. We preferred this method over a screening using the mean and standard deviation due to its superior robustness with respect to larger outliers.

Individual biases outside the interval $\langle \text{median}[b_i(t,\phi,z)] \pm 10 \cdot \text{MAD}[b_i(t,\phi,z)] \rangle$, with $i = 1, ..., n_c(t,\phi,z)$, were discarded. For a normally distributed set of data $10 \cdot \text{MAD}$ correspond roughly to 7.5 standard deviations. Hence this has not been a very strict screening, aiming to remove the most prominent outliers of individual biases $b_i(t,\phi,z)$.

As indicated by Eqs. 6 – 8 the biases were calculated for various sets of coincidences covering different times $t$ and latitude bands $\phi$ as listed below:

• time $t$: MAM, JJA, SON, DJF and all seasons together

• latitude $\phi$: 90°S – 60°S (also referred to as Antarctic), 60°S – 30°S, 30°S – Equator, 15°S – 15°N (also referred to as tropics), Equator – 30°N, 30°N – 60°N, 60°N – 90°N (also referred to as Arctic) and 90°S – 90°N (also referred to as global)

The comparisons were performed on pressure as altitude scale and biases have been derived in the volume mixing ratio space.

For this all data sets were interpolated on a common grid with 32 levels per pressure decade. Tropospheric data were intentionally removed using MERRA tropopause information. Comparisons in the troposphere will be presented by Read et al. (in preparation). Due to the finite vertical resolution of the individual data sets the removal of tropospheric data has not been perfect and at the lower boundary volume mixing ratios still remain that are associated with tropospheric conditions. In comparisons where differences in the vertical resolution among the data sets had to be considered the tropospheric data were removed after

the convolution to obtain optimal results. In the following figures we show only bias results that are based on at least 20 coincidences to avoid spurious results. This targets primarily the lower and upper vertical limits of the comparisons, where typically the smallest numbers of coincidences tend to occur.

Given the large number of data sets, this work yields a large number of comparisons. Even though every comparison is unique some sort of combination is needed to be able to present the results in a reasonable way. To summarise the bias results

for a given data set considering a specific time and latitude band we chose the median over all available comparisons (with an aggregation of the MIPAS results as described later in Sect. 3.5). We tested other approaches but the median appeared to be the optimal choice for multiple reasons. It provides robust statistics in the presence of outliers (avoiding the need for additional screening) and it does not require any assumption of a certain probability distribution nor a specific weighting of the individual comparisons.



### 3.4 Drift analysis

Besides the bias estimation we also performed an analysis of drifts among the different data sets. Unlike for the bias comparisons, we do not separate the drift comparisons by season. The drift analysis was based on monthly averaged biases derived from a minimum of 5 coincidences. Drifts were only calculated if the overlap period between the two data sets compared was

at least 36 months. This period is defined as the time between the first and the last month where sufficient coincidences were found between the two data sets. The estimation of the drifts was done with a regression model that contained an offset, a single linear term for the drift as well as terms for the semi-annual (SAO), annual (AO) and quasi-biennial oscillation (QBO):

$$
\begin{aligned}
f(t,\phi,z) = & C_{\text{offset}}(\phi,z) + C_{\text{drift}}(\phi,z) \cdot t \\
& C_{\text{SAO}_1}(\phi,z) \cdot \sin(2 \cdot \pi \cdot t / p_{\text{SAO}}) + \\
& C_{\text{SAO}_2}(\phi,z) \cdot \cos(2 \cdot \pi \cdot t / p_{\text{SAO}}) + \\
& C_{\text{AO}_1}(\phi,z) \cdot \sin(2 \cdot \pi \cdot t / p_{\text{AO}}) + \\
& C_{\text{AO}_2}(\phi,z) \cdot \cos(2 \cdot \pi \cdot t / p_{\text{AO}}) + \\
& C_{\text{QBO}_1}(\phi,z) \cdot QBO_1(t) + \\
& C_{\text{QBO}_2}(\phi,z) \cdot QBO_2(t).
\end{aligned}
\tag{9}
$$

In the equation $f(t,\phi,z)$ denotes the fit of the regressed bias time series. $C$ are the regression coefficients of the individual

model components and $C_{\text{drift}}$ describes the drift that is sought. The SAO and AO are parameterised by orthogonal sine and cosine functions, while for the QBO the normalised winds at $50\,\text{hPa}$ ($QBO_1$) and $30\,\text{hPa}$ ($QBO_2$) observed over Singapore (1°S, 104 °E) are used. These winds are closely orthogonal and have been compiled by Freie Universität Berlin (webpage: http://www.geo.fu-berlin.de/met/ag/strat/produkte/qbo/qbo.dat). $p_{\text{SAO}}$ and $p_{\text{AO}}$ represent the time periods of the semi-annual (0.5 years) and annual variation (1 year), respectively. The regression coefficients were derived following the method by von

Clarmann et al. (2010) using the standard mean error of the monthly averaged biases as statistical weights. In the regression also autocorrelation effects and empirical errors are considered, using the same approach as outlined by Stiller et al. (2012b).

### 3.5 Aggregation of the MIPAS results

The previous WAVAS-II papers (Lossow et al., 2017; Nedoluha et al., 2017; Khosrawi et al., 2018) often received comments on the large number of MIPAS data sets (here 13 out of 33, see Tab. 1 for example) included in the assessment. As described in

these publications the different MIPAS data sets are based on different measurement modes (with different vertical sampling) and, more prominently, are derived by four different processors with varying retrieval choices, as microwindows, vertical grid regularisation, spectroscopic database or a priori for example. Here, we want to provide general results in the form of percentiles and histograms using all comparison results as well as a summaries of data set specific biases as described at the last paragraph of Sect. 3.3. In general such results will always depend on the data sets that are considered. One of the WAVAS-

II goals was to involve as many data sets as possible to provide a rather complete and realistic picture. There are however





limits. For example, if all data sets in such assessment would be experimental (i.e. test or research versions), any general result derived from the combination of them would be rather meaningless. Also the large number of MIPAS data sets in our assessment may be such a limit. Accordingly, we asked ourselves if our intended results may be influenced or skewed by the large number of MIPAS data sets. The typical biases among the different MIPAS data sets are significantly smaller than among

the non-MIPAS data sets. They amount to roughly 0.1 ppmv (0.5 ppmv) for the MIPAS (non-MIPAS) data sets, considering large parts of the stratosphere. A similar picture is found in terms of typical drifts. In the stratosphere they are approximately $0.1 \, \mathrm{ppmv} \cdot \mathrm{decade}^{-1}$ for the MIPAS data sets while for the non-MIPAS data sets they correspond to $0.3 \, \mathrm{ppmv} \cdot \mathrm{decade}^{-1}$. This indicates a relative similarity among the different MIPAS data sets in contrast to the non-MIPAS data sets. This clearly can affect our intended general results based on all comparison results. In addition, the summary biases (based on the median

over all comparisons to the other data sets, see Sect. 3.3) for any randomly picked MIPAS data set will be small because this data set is compared to many relatively similar ones. In contrast, a single non-MIPAS data set has to compare to the bulk of MIPAS data sets. If these comparisons disagree, the summary biases for this non-MIPAS data set will be large. Given these considerations we decided to aggregate the MIPAS results. For percentiles, histograms and full matrix plots the aggregation has been performed as follows:

- all MIPAS comparison results to a given non-MIPAS data set are combined using the median

   - comparison results between different MIPAS data sets are not considered any further

For the summary bias $\overline{b}_S(t, \phi, z)$ of a given data set, described in Sect. 3.3 and shown in Figs. 8 and 9 in the main manuscript as well as in Fig. S8 in the Supplement, the following approach has been chosen:

(a) for a non-MIPAS data set, like HALOE

$$\overline{b}_S(t, \phi, z, \mathrm{HALOE}) = \mathrm{median}\langle \overline{b}(t, \phi, z, ds_1), \mathrm{median}[\overline{b}(t, \phi, z, ds_2)]\rangle \tag{10}$$

   where $\overline{b}(t, \phi, z, ds_1)$ represents all biases of the HALOE data set to the remaining non-MIPAS data sets and $\overline{b}(t, \phi, z, ds_2)$ describes the HALOE biases relative to all MIPAS data sets

(b) for a given MIPAS data set, like MIPAS-Bologna V5R NOM

   $$\overline{b}_S(t, \phi, z, \mathrm{MIPAS - Bologna \ V5R \ NOM}) = \mathrm{median}\langle \overline{b}(t, \phi, z, ds_3), \mathrm{median}[\overline{b}(t, \phi, z, ds_4)]\rangle \tag{11}$$

where $\overline{b}(t, \phi, z, ds_3)$ are all the biases to non-MIPAS data sets and $\overline{b}(t, \phi, z, ds_4)$ represents the biases of the MIPAS-Bologna V5R NOM data set to the remaining MIPAS data sets

In numerous figure we supply as auxiliary information the number of comparisons or data points contributing to the results presented. Even, when the MIPAS results are aggregated we still count the contributing results individually and do not condense them into a single contribution. For example in Fig. 9 the bias summaries for the ACE-FTS v3.5 data set are presented. This





data set could be compared to all 13 MIPAS data sets if coincidences at all seasons and latitudes are considered. Hence the number of comparisons contributing to these summary biases given in that figure (i.e. 31), includes these 13 comparisons.

We will show some results with and without the aggregation of the MIPAS results for the sake of comparison. In the main manuscript this concerns Figs. 4, 5 and 11. In the Supplement Figs. S1, S4 and S9 show percentiles and histograms without

the aggregation of the MIPAS results that correspond respectively to Figs. 6, 7 and 12 in the main manuscript which take this aggregation into account. The two ACE-FTS and SCIAMACHY solar occultation data sets are also based on the same set of measurements. Therefore also an aggregation of these results could be considered. However, due to the small number of the data sets concerned (in relation to the MIPAS data sets), this has not further pursued.

## 4 Bias results

The presentation of the bias results is split into three parts. We start with an example to provide a first impression of the analyses. Then, we focus on a general, data set independent, assessment of the biases. This aims to provide a picture of the typical bias characteristics found in the observational database. In the last part of this section, specific results for individual data sets are presented.

The upper triangle of Fig. 2 provides a quick overview which data sets were compared in terms of biases for any of the

time-latitude bins considered (see Sect. 3.3). The presentation uses a traffic light system:

(1) green: Comparisons were performed.

(2) yellow: Comparisons were performed. However, the minimum criterion of at least 20 coincidences (as defined in Sect. 3.3) was not met at any considered altitude. This concerns four comparisons, namely the comparisons of the HALOE and SAGE II data sets to SCIAMACHY lunar data set as well as and the comparisons of the MAESTRO data set to both

SMILES data sets.

(3) red: No comparison could be performed as the data sets do not overlap.

Complementary to this, Fig. 3 shows the number of coincident observations among the data sets (considering all seasons and latitude bands). The HIRDLS and MLS data sets yield more than 3 million coincidences according to our criteria, the largest number found in our comparisons. The comparisons among the different MIPAS V5R NOM data sets comprise more

than 1.7 million coincidences. On the opposite end, less than 100 coincident observations are found in the comparisons of the following data sets: ACE-FTS vs. SMILES, GOMOS vs. SCIAMACHY occultation (both lunar and solar), GOMOS vs. SMILES, HALOE vs. MIPAS V5R MA, ILAS vs. SCIAMACHY lunar, MAESTRO vs. SCIAMACHY lunar as well as SAGE II vs. SCIAMACHY lunar.

### 4.1 Example

Figure 4 shows exemplarily biases of the SCIAMACHY solar OEM data set, considering coincident observations during all seasons and at all latitudes. The upper row considers biases in absolute terms, while the lower row focuses on biases in relative





terms. The left panels show the biases to the individual data sets (i.e. SCIAMACHY solar OEM minus the other data set, see Sect. 3.3). In the right panels the corresponding summary biases are shown. The red profile is based on the median over all comparisons (see Sect. 3.3). The blue profile, additionally, considers the aggregation of MIPAS results as described in Sect. 3.5 and is also used for the summary of the data set specific results presented later in Sect. 4.3 and the Supplement. The legend

provides information on the actual temporal and spatial coverage of the individual comparisons as a complement. Even though all latitudes are considered in the analysis, the comparisons are limited to the latitude range between $49°$N and $69°$N according to the coverage of the SCIAMACHY solar OEM data set.

The comparisons indicate biases of the SCIAMACHY solar OEM data set, that are typically within $\pm 1$ ppmv or $\pm 20\%$ in relative terms. In most cases the biases are positive, but in some comparisons also negative biases are found. These negative

biases are visible in the lower (roughly between $100$ hPa and $50$ hPa) and upper stratosphere (between $3$ hPa and $1$ hPa) as well as in the lower mesosphere (roughly above $0.2$ hPa). In the uppermost altitude range this behaviour is systematically observed in comparisons to the MIPAS Bologna data sets derived from the nominal mode observations, i.e. MIPAS-Bologna V5H and MIPAS-Bologna V5R NOM. For the other altitude ranges no such data set specific behaviour is observed. Beyond that, these example biases indicate more issues with specific data sets that will be presented more comprehensively in Sect. 4.3 and the

Supplement.

In accordance with the individual bias results presented in the left panels, the summary profiles shown in the right panels indicate generally positive biases for the SCIAMACHY solar OEM data set compared to the other data sets. From the summary biases we find that the results are clearly influenced by the summary approach in the altitude range between $30$ hPa and $0.6$ hPa. Here, the median over all comparisons yields consistently lower biases than the median considering the aggregation

of the MIPAS results. Differences between these two profiles become as large as $0.4$ ppmv, corresponding to $6\%$ in relative terms. This highlights the influence that the large number of MIPAS data sets can have in the comparisons, as discussed in Sect. 3.5. The summary profiles considering the aggregation of the MIPAS results exhibit the smallest biases below $25$ hPa (around $0.25$ ppmv or $5\%$) and at $0.1$ hPa (about $0.1$ hPa or $2\% - 3\%$). At $10$ hPa and more prominently at $0.25$ hPa the biases maximise. At $10$ hPa the bias amounts to $0.75$ ppmv or $12\%$ while the maximum in the lower mesosphere (also notable in the

summary profiles without aggregation) exhibits smaller values ($0.6$ ppmv or $10\%$). On average, the biases amount to $0.5$ ppmv (about $8\%$) in the stratosphere.

## 4.2   General results

The left column of Fig. 5 shows the biases from the full matrix of comparisons. Here, the comparisons that include coincident observations during all seasons and at all latitudes are considered. The upper panel shows the results for the absolute biases,

in the lower panel the results for the relative biases are given. Based on our comparison approach (see Sect. 3.1) the results for the full matrix are symmetric around zero. In grey the comparison results without the aggregation of the MIPAS results are shown. With 33 data sets theoretically $33 \cdot 32 = 1056$ comparisons (of which 528 are unique) are possible. But since not all data sets overlap with each other the actual number decreases to 862 comparisons (of which 431 are unique, see Fig. 2). For 8 comparisons (4 unique) the biases are based on less than 20 coincidences at all altitudes and were thus not



considered any further (see Sect. 3.3 or description of Fig. 2 in the beginning of this section). Hence, the unaggregated results are effectively based on 854 comparisons (427 unique). In blue the comparison results considering the aggregation of the MIPAS results are shown. As described in Sect. 3.5 the aggregation omits comparisons among the MIPAS data sets, reducing the amount of available comparisons to 770. After combining all MIPAS results in comparisons to non-MIPAS data sets finally

348 comparisons remain for the full matrix.

Overall, the left column of Fig. 5 provides a good first impression of the typical envelope of biases in the observational database. Above 30 hPa the biases are typically within $\pm 2$ ppmv (or $\pm 40\%$). Below this altitude the biases can get significantly larger and even exceed $\pm 5$ ppmv or $\pm 100\%$ in some occasions.

Based on the positive biases shown in the left panels of Fig. 5 the right panels show the corresponding 50% (i.e. median,

blue), 80% (green) and 95% (red) percentiles without (lighter colours) and with (darker colours) the aggregation of the MIPAS results. In general, the 50% and 80% percentiles are quite constant above 70 hPa, while the 95% percentile shows much more variation in this altitude range. At altitudes below there is a distinct increase in the corresponding values. In addition, the percentiles considering the aggregation of the MIPAS results are larger than without this aggregation. At stratospheric altitudes the differences amount to 0.1 ppmv (2%) for the 50% percentile, 0.2 ppmv (5%) for the 80% percentile and 0.5 ppmv (7%)

for the 95% percentile, respectively. Prominent exceptions from this behaviour are observed close 0.1 hPa, below 200 hPa (except for 50% percentile of the absolute biases) or between 2 hPa and 1 hPa for the 95% percentile of the absolute biases. In the following description we focus on percentiles considering the aggregation of the MIPAS results. The 50% percentile is around 0.5 ppmv above 100 hPa and minimises at 60 hPa with 0.35 ppmv. Below 200 hPa the 50% percentile exceeds 2 ppmv. In relative terms, the 50% percentile is smaller than 10% around 60 hPa and between 25 hPa and 0.3 hPa. From

100 hPa to 250 hPa the 50% percentile increases from 12% to 40%. Below, the percentile actually decreases again to reach a pronounced minimum of 23% at about 340 hPa. The 80% percentile considering the absolute biases averages to 1.1 ppmv for altitudes above 100 hPa. A distinct minimum is again observed at 60 hPa (0.8 ppmv). Also, in the altitude range between 10 hPa and 3 hPa as well as around 0.5 hPa and 0.1 hPa pronounced minima of about 1 ppmv are visible. At 100 hPa the 80% percentile amounts to 1.5 ppmv. With decreasing altitude it quickly increases and exceeds 5 ppmv slightly below 200 hPa. For

the relative biases the 80% percentile varies between 20% and 35% in the altitude range between 100 hPa and 10 hPa. Higher up, it is below 20% with a few exceptions. Below 200 hPa the 80% percentile ranges from 50% to 70%. Again a pronounced minimum is observed close to 370 hPa, similar as observed for the 50% percentile. The 95% percentile is generally smaller than 2 ppmv with three noticeable exceptions. One concerns the altitude range below 70 hPa, in a similar fashion as observed for the other two percentiles. Another exception is observed around 30 hPa where a localised maximum of more than 5 ppmv

is found. This behaviour can be attributed to the MAESTRO data set that is close to its upper boundary and exhibits high positive biases at these altitudes. The third exception is visible at 0.7 hPa, primarily attributed to the two SAGE data sets. In relative terms, the 95% percentile ranges from 20% to almost 80% above 70 hPa. Below 100 hPa there are large variations and the values exceed 100% occasionally.

In Fig. 6 the characterisation of the typical biases is extended by considering the 50% percentile (median) for different

seasons and latitude bands. These results take into account the aggregation of the MIPAS results and are again based on the





positive biases only, as the percentile results shown in the previous figure. The left column of the figure considers the absolute biases, the right column focuses on the relative biases. The different rows focus on different seasons or their combination. The results for the different latitude bands are colour-coded. On the right side of the individual panels the number of (unique) comparisons contributing to results are given. As described in Sect. 3.5 the comparisons to the different MIPAS data sets are
counted individually. In general, the 50% percentiles exhibit a rather common altitude dependence for the different seasons and latitude bands. Below about 70 hPa the 50% percentiles increase considerably and the highest values are observed in the tropics and subtropics. Below 200 hPa the values are typically beyond the upper limits of the x-axes considered here, i.e. 1 ppmv in absolute terms and 20% in relative terms. The 50% percentiles are typically lowest in the altitude range from roughly 70 hPa to 5 hPa. The values here vary between 0.25 ppmv and 0.5 ppmv in absolute terms and between 5% and 10% in relative terms.
In this altitude region, the lowest values generally occur outside the polar regions. Higher up, there is a distinct increase of the 50% percentiles within a small altitude range, i.e up to about 1 hPa. At this altitude the 50% varies approximately between 0.6 ppmv and 0.8 ppmv (roughly 8% to 12%). In JJA the values are a bit smaller while in DJF there is a much larger variation among the latitude bands (percentiles minimise for the Arctic and maximise for the latitude band from 30°N and 60°N). Higher up, the 50% percentiles vary considerably with altitude and among the latitude bands, comprising values from 0.3 ppmv to
1 ppmv (4% to 20%). The smallest values are typically observed between 0.5 hPa and 0.4 hPa. In MAM, SON and all seasons combined the latitude band from the Equator to 30°N stands for these minimum values, in DJF this occurs prominently in the Arctic. The largest values are observed at 0.1 hPa, with pronounced variations among seasons and latitude bands.

Figure S1 in the Supplement shows the results corresponding to Fig. 6 without the aggregation of the MIPAS results. Overall, the altitude dependence is quite similar to the results shown here. However, without the aggregation, the values for the 50%
percentiles are smaller (like in Fig. 5), as is the variation among the different latitude bands and seasons (a prominent exception occurs at 0.1 hPa). To further complement Fig. 6 the results for the 80% and 95% percentiles, considering again the aggregation of the MIPAS results, are shown in the Supplement (Figs. S2 and S3). For these larger percentiles the altitude dependence is somewhat different, in particular for the 95% percentile where less pronounced differences between stratospheric and lower mesospheric values are visible. Pronounced differences among the latitude bands rather occur in the lower stratosphere than
lower mesosphere.

For a last characterisation of the biases in the observational database we use histograms, as shown in Fig. 7. These results use again the positive biases only, consider data from all altitudes and take into account the aggregation of the MIPAS results. The histograms for the absolute biases (left column) use bins of 0.05 ppmv. For the relative biases (right column) bins of 1% are considered. As in the previous figure the different panels consider different seasons or their combination. The different latitude
bands are again colour-coded. On the right side of the individual panels the number of data points contributing the results are given (again comparisons to different MIPAS data sets are counted individually, see Sect. 3.5). Overall, the histograms exhibit a rather similar picture for the different seasons and latitude bands. Their shapes are quite close to a Gaussian distribution. Typically, between 5% and 7% of the absolute biases are found within the first bin that ranges from 0 ppmv to 0.05 ppmv. The decrease in occurrence towards larger biases (up to 0.8 ppmv) is steepest in DJF. For biases around 1 ppmv the occurrence has
dropped to 1% to 2.5%. Biases beyond 3 ppmv occur in 2% to 6% of the comparisons. The lowest occurrences for these biases





are observed in the Antarctic and tropics. In relative terms the occurrence of biases within the first bin from 0% and 1% varies between 4% and 7% depending on season and latitude band. A bias of 10% occurs in 3% to 5% of the comparisons. For biases beyond 50% the occurrence is between slightly less than 2% and 9%. The lower limit is observed for the latitude band between 30°S and the Equator in JJA. In contrast, the upper limit occurs in the Antarctic in DJF.

As for the previous figure we present the results without the aggregation of the MIPAS results, corresponding to Fig. 7, in the Supplement (Fig. S4). The results without the aggregation exhibit a larger occurrence rate for the smallest biases (between 7% and 10%). Besides that, the variation among the different latitude bands is smaller. In the Supplement we present in addition histograms that focus separately on data in the altitude ranges $100\,\text{hPa} - 10\,\text{hPa}$, $10\,\text{hPa} - 1\,\text{hPa}$ and $1\,\text{hPa} - 0.1\,\text{hPa}$ (see Figs. S5, S6 and S7), again considering the aggregation of the MIPAS results. From these figures we find that for small biases

(i.e. up $0.5\,\text{ppmv}$ or 10%) the occurrence rate is smallest in the altitude range from $1\,\text{hPa}$ to $0.1\,\text{hPa}$. For relative biases the highest occurrence of such biases is observed in the altitude range from $10\,\text{hPa}$ to $1\,\text{hPa}$. In contrast, large biases ($>3\,\text{ppmv}$ or 50%) occur by far most often in the altitude range from $100\,\text{hPa}$ to $10\,\text{hPa}$. The results for the altitude range from $10\,\text{hPa}$ to $1\,\text{hPa}$ deviate to some extent from a Gaussian distribution. In particular, the tropics and subtropics show the maximum occurrence rates for biases of $0.4\,\text{ppmv}$ (7% to 8%). Also, the variations among the latitude bands increase with altitude.

Finally, we want to note that the consideration of differences in the vertical resolution among the data sets, as done in this work (see Sect. 3.2), yields an improvement of the biases in 55% of the comparisons (all data sets, times, latitude bands and altitudes). This primarily concerns altitudes above $70\,\text{hPa}$ and pronounced improvements are visible around $30\,\text{hPa}$, the stratopause and above $0.2\,\text{hPa}$. They can reach several tenths of a ppmv (or a few percent) in these altitude regions. Below $70\,\text{hPa}$ there are both improvements and deteriorations, however with a clear tendency to the latter and as large as 100% in

relative terms. This indicates that differences in the vertical resolution are not the primary cause of the pronounced biases in this altitude region.

## 4.3   Data set specific results

In this section an overview of data set specific results is presented. Figure 8 shows the summary biases for all data sets, based on the comparisons that consider coincidences during all seasons and at all latitudes. The absolute biases are given in the left

column and the relative biases in the columns. For the sake of better visibility the results have been split into three panels. In the legend this separation is also reflected by the columns. In addition the legend contains the information on the number of comparisons contributing the individual summary biases. As described in Sect. 3.5 these results consider the aggregation of the MIPAS results. Overall the smallest biases are observed in the middle and upper stratosphere. The largest biases occur below $100\,\text{hPa}$ (larger than $\pm2\,\text{ppmv}$ or $\pm50\%$). On occasion also pronounced summary biases are visible in the lower mesosphere

close to $0.1\,\text{hPa}$.

    The first row comprises the results for the ACE-FTS, GOMOS, HALOE, HIRDLS, ILAS-II, MAESTRO and some MIPAS data sets (Bologna and one from ESA). Here, the biases are relatively small with $\pm0.4\,\text{ppmv}$, corresponding to a relative biases of $\pm10\%$. Larger (negative) biases are found for the GOMOS, HALOE (above $2\,\text{hPa}$) and ILAS-II data sets. Among them, the GOMOS data set shows the absolute largest biases. The GOMOS biases get more negative with increasing altitude above



75 hPa and reach values of $-1.4$ ppmv at 20 hPa. Up to 2 hPa the biases of the GOMOS data set vary between $-1.7$ ppmv and $-1.1$ ppmv (roughly $-35\%$ to $-25\%$). Above 2 hPa the biases decrease again in size, however this is close to the upper limit where water vapour information can be retrieved from the GOMOS observations. The biases for the ILAS-II data set generally do not exceed $-0.8$ ppmv, corresponding to less than $-15\%$ in relative terms. For the HALOE data sets the biases

vary between $-0.5$ ppmv and $-0.2$ ppmv ($-5\%$ to $-10\%$) in the altitude range from 70 hPa to 5 hPa. Towards the lower mesosphere the biases increase in size, where they are around $-1$ ppmv ($-15\%$). At 0.1 hPa the biases for the data sets shown in the first panel range from $-0.9$ ppmv ($-20\%$) to 0.4 ppmv (10%). Below 100 hPa the absolute biases increase significantly in size. All data sets exhibit biases larger than $\pm 2$ ppmv at some altitude. Also, in relative terms a large variation is observed, however the biases for ACE-FTS v2.2 and MAESTRO data sets remain largely within $\pm 10\%$.

In the second panel results for numerous MIPAS (ESA, IMKIAA and Oxford) data sets are shown, plus those from the MLS and POAM III data sets. For these data sets the biases are generally within $-0.3$ ppmv to 0.6 ppmv, corresponding roughly to relative biases between $-5\%$ and 10%. Larger biases are found for the MIPAS-ESA V5R MA, MIPAS-Oxford V5H and MIPAS-Oxford V5R MA data sets around the stratopause. Some data sets exhibit a pronounced increase in their biases above 0.3 hPa. This concerns the MIPAS-Oxford and the MLS data sets. At 0.1 hPa biases range, overall, from $-0.4$ to more than

2 ppmv, corresponding to relative biases of $-10$ to 45%. Below 100 hPa again large biases are visible, with a tendency towards negative values. Data sets for which the biases exceed $-50\%$ below 100 hPa are MIPAS-ESA V7R, MIPAS-Oxford V5H, MIPAS-Oxford V5R NOM. The MIPAS-ESA V5R NOM ($-25\%$ to 0%), MIPAS-IMKIAA ($-20\%$ to 5%), MLS ($\pm 20\%$) and POAM II (5% to 15%) data sets show here typically smaller biases.

The third panel of Fig. 8 presents results for data sets from the following instruments: SAGE, SCIAMACHY, SMILES,

SMR and SOFIE. Here, some data sets exhibit quite pronounced biases. This concerns on one hand the experimental SMILES data sets. They cover the altitude range from slightly above 200 hPa to 50 hPa and show good agreement between 70 hPa and 60 hPa. Higher up, distinct positive biases (exceeding 1 ppmv or 20%) are observed while below negative biases of even larger size are visible. The summary biases for the SMR 544 GHz data set are almost entirely negative. Around 30 hPa they amount to $-1.8$ ppmv (exceeding -50%). Above 15 hPa they get significantly smaller and switch sign at above 10 hPa, close

to the upper limit of this data set. The biases for the SMR 489 GHz data set are quite low up to 10 hPa, but start to increase significantly higher up. In the lower mesosphere this data set exhibits biases around $-1.2$ ppmv in absolute terms and $-25\%$ in relative terms. In the lower half of the stratosphere the SAGE II, SAGE III, SCIAMACHY lunar and SOFIE data sets show very good agreement (typically within $\pm 0.2$ ppmv or $\pm 5\%$). Towards higher altitudes biases increases to some extent, most prominently for the SAGE II data set above 3 hPa. The SCIAMACHY solar occultation data sets exhibit low biases in the

lower stratosphere (around 5%). Above 30 hPa they vary typically between 0.3 ppmv and 0.7 ppmv (5% to 12%).

Figure 9 shows the summary biases for the ACE-FTS v3.5 data set as function of season and latitude, using the same layout as in Figs. 6 and 7. For all other data sets the corresponding figures are provided in the Supplement (Fig. S8). The comparisons to the ACE-FTS v3.5 data set show a rather consistent picture above about 100 hPa with relatively small variations among the different seasons and latitude bands. Between about 80 hPa and 5 hPa the biases are typically within $\pm 0.2$ ppmv or $\pm 5\%$, with

a clear preference towards negative biases. Towards higher altitudes the biases get more negative. They peak between 2 hPa



and 1 hPa with values from $-0.8$ ppmv ($-10\%$) to $-0.2$ ($-2\%$), depending on season and latitude. Above 1 hPa, the biases decrease again and switch sign at around 0.4 hPa to 0.3 hPa. At 0.1 hPa there is a larger bias variation with season and latitude band. Here, the biases range between 0 ppmv and 0.9 ppmv (0% to 15%). Below 100 hPa a wide range of biases is observed, occasionally they exceed $\pm 2$ ppmv. The vast majority of the biases are negative. In relative terms they vary roughly between -40% and 10%.

## 5 Drift results

For the presentation of the drift results we choose a similar approach as for the bias results. First an example is shown, followed by a general assessment of the drifts in the observational database. After that we present data set specific drift results. Finally, we compare the drift results to those obtained from the comparisons of monthly zonal mean time series presented in the work by Khosrawi et al. (2018). This aims to quantify how dependent the drift results are upon the actual method to derive them.

As for the biases, the lower triangle of Fig. 2 provides an overview for which data set combinations drift comparisons were possible for any of the time-latitude bins considered in this work. In this context the yellow colour means that the overlap criterion of at least 36 months was not met and thus the drift results were not considered any further (see Sect 3.4). The overlap periods among the different data sets are shown in the lower triangle of Fig. 3. These numbers are based on the comparisons considering coincidences during all seasons and at all latitudes, maximising these periods. The longest overlap period is found between the two SMR data sets and amounts to 153 months. The comparisons of the SMR data sets to the ACE-FTS v3.5, MAESTRO and MLS data sets yield overlap periods periods beyond 120 months. The same is true for the comparisons of the ACE-FTS v3.5 data set to the MAESTRO and MLS data sets. Contrary, overlap periods of less than 40 months are on one hand found in the comparisons to the HIRDLS data set, which itself only comprises 39 months of data. On the other hand, the following comparisons have overlap periods between 36 months and 39 months: GOMOS vs. HALOE, GOMOS vs. SCIAMACHY lunar, HALOE vs. SCIAMACHY limb, POAM III vs. SCIAMACHY solar occultation, SAGE II vs. SCIAMACHY limb and SAGE III vs. SCIAMACHY limb.

### 5.1 Example

As example we consider the drift of the SMR 489 GHz data relative to other data sets in the latitude range from 30°N to 60°N. The drift estimates are shown in the left panel of Fig. 10. The right panel shows the corresponding significance levels, defined as the absolute ratio between the drift estimates and their associated uncertainties. In the legend for every data set two numbers are provided. The first number indicates the overlap period of the two data sets in months. As described in Sect. 3.4, a minimum overlap period of 36 months was required for a drift to be calculated. The second number shows during how many months the two data sets actually have a sufficient number of coincidences. Since this information is altitude-dependent the legend considers the maximum values over all altitudes. The example indicates mostly positive drifts for the SMR 489 GHz data set, which means that its trends in water vapour are more positive or less negative than the trend estimates derived from the other data sets. The drifts are clearly systematic. Even though the comparisons consider different time periods, and thus





the estimates can vary, a very consistent picture of their altitude dependence is obtained. Pronounced drifts are observed around 50 hPa, which is close to the lower altitude limit where water vapour retrievals from SMR observations are possible. The drifts are as large as $2\,\mathrm{ppmv}\cdot\mathrm{decade}^{-1}$ and many of those are also statistically significant at the $2\sigma$ uncertainty level. Towards 20 hPa the drift estimates decrease to values smaller than $0.5\,\mathrm{ppmv}\cdot\mathrm{decade}^{-1}$. The comparisons to a few data sets

even indicate negative drift estimates for the SMR 489 GHz data set. Above 20 hPa, the drifts increase again and maximise at around 0.5 hPa. Here, the drift estimates typically range from $1\,\mathrm{ppmv}\cdot\mathrm{decade}^{-1}$ to $1.75\,\mathrm{ppmv}\cdot\mathrm{decade}^{-1}$ and are in most cases statistically significant. An exception are HALOE and SAGE II for which drift estimates are even larger than 2 ppmv. Above 0.5 hPa the drift estimates generally decrease again.

## 5.2   General results

Figure 11 shows the drift estimates from the full matrix of comparisons, similar to the bias results shown in Fig. 5. Again, these results consider the comparisons that incorporate coincidences during all seasons and at all latitudes. The upper panel shows the picture without the aggregation of the MIPAS results, the picture in the lower panel takes this aggregation into account. Overall, from the 862 comparisons (see Sect. 4.2) in the full matrix 470 comparisons yield drift results (see Fig. 2), with the chosen minimum overlap period of 36 months. In 450 comparisons drift estimates that are statistically significant at the $2\sigma$ uncertainty

level are found. The estimates that are significant are marked in light blue in the figure. The picture without the aggregation of the MIPAS results indicates a wide range of drifts, in particular below 30 hPa and at 0.1 hPa. Some of the extreme values can be assigned to specific data sets. Between 30 hPa and 10 hPa the envelope of the drifts is smallest. Here, they are generally within $\pm0.4\,\mathrm{ppmv}\cdot\mathrm{decade}^{-1}$. Also median-wise, in this altitude range the smallest drifts are observed. Beyond that, we find that there is some dependence between the overlap period and the absolute drift size. Based on results above 100 hPa, the drift

size typically decreases with increasing overlap period for periods up to 70 months. Beyond that overlap period, there is no obvious connection.

The picture with the aggregation of the MIPAS results is clearly sparser than that without the aggregation. Mostly prominently this is visible in the lower mesosphere, similar to the corresponding picture for the biases shown in Fig. 5. In contrast to the unaggregated picture, a notable widening of the drift range at 0.1 hPa is not observed. At lower altitudes the picture is rather

similar and the envelope of drifts exhibits a similar minimum region in the middle stratosphere. The typical drift sizes (based on the median) are typically larger for the picture with the aggregation of the MIPAS results, except above 0.2 hPa. Below 30 hPa the difference is of the order of $0.3\,\mathrm{ppmv}\cdot\mathrm{decade}^{-1}$. Above 20 hPa the difference is roughly $0.1\,\mathrm{ppmv}\cdot\mathrm{decade}^{-1}$.

In Fig. 12 drift histograms are shown for the different latitude bands, taking into account the aggregation of the MIPAS results. They consider the positive estimates from all altitudes that are statistically significant at the $2\sigma$ uncertainty level.

In the upper right corner of the figure the number of available data points is indicated. Bins of $0.05\,\mathrm{ppmv}\cdot\mathrm{decade}^{-1}$ are used for the histograms. The smallest drifts occur in less than 1% of comparisons, consistently for all latitudes. Typically such small drifts are not statistically significant. Correspondingly the histograms do not resemble a Gaussian distribution, as typically found for the biases. Considering both statistically significant and non-significant results a more Gaussian-like picture is found (not shown here). Beyond the smallest drift bin, the occurrence rates quickly rise. In fact, the maximum





occurrence rates are observed for drifts between $0.05\,\mathrm{ppmv}\cdot\mathrm{decade}^{-1}$ and $0.2\,\mathrm{ppmv}\cdot\mathrm{decade}^{-1}$ and vary between 7.5% and 9.5%. In particular, for the latitude bands from 15°S to 15°N and from the Equator to 30°N this maximum already occurs in the bin from $0.05\,\mathrm{ppmv}\cdot\mathrm{decade}^{-1}$ to $0.1\,\mathrm{ppmv}\cdot\mathrm{decade}^{-1}$. Towards larger drifts the occurrence rates generally decrease. However there are some prominent additional maxima, for example in the Antarctic for drifts between $0.45\,\mathrm{ppmv}\cdot\mathrm{decade}^{-1}$

and $0.5\,\mathrm{ppmv}\cdot\mathrm{decade}^{-1}$ (occurrence rate of more than 7%). For drifts of $1\,\mathrm{ppmv}\cdot\mathrm{decade}^{-1}$ the occurrence rate is between 1% and 2%, except for the latitude bands from 15°S to 15°N and from the Equator to 30°N. In those latitude bands again pronounced additional maximum in the occurrence rate (between 3% and 4%) are visible. For drifts of $2\,\mathrm{ppmv}\cdot\mathrm{decade}^{-1}$ the occurrence rate is smaller than 1% and minimises for the latitude band from the Equator to 30°N, by a small margin compared to the Antarctic. Drifts larger $3\,\mathrm{ppmv}\cdot\mathrm{decade}^{-1}$ occur in 2% to 7.5% of the comparisons with the highest occurrence rate for

the latitude band from 60°to 30°S and the lowest occurrence rate for the latitude band between 15°S and 15°N.

In the Supplement in Fig. S9 the corresponding picture without the aggregation of the MIPAS results is shown. In contrast to Fig. 12, the maximum occurrence rates are a bit larger (between 8% and 10%). For the Antarctic and Arctic the maximum occurrence rates just occur for drifts between $0.3\,\mathrm{ppmv}\cdot\mathrm{decade}^{-1}$ and $0.35\,\mathrm{ppmv}\cdot\mathrm{decade}^{-1}$. For a number of latitude bands a double peak structure is visible, so that the drift range with large occurrence rates (i.e. $>5\%$) is much wider on one hand.

On the other hand, the occurrence rates for drifts larger than $1\,\mathrm{ppmv}\cdot\mathrm{decade}^{-1}$ are much smaller than for the picture with the aggregation of the MIPAS results. Also, the variation with increasing drifts and among the latitude bands is smaller in Fig. S9.

## 5.3   Data set specific results

In this section we provide summaries of data set specific drift results, focusing on the MIPAS-ESA V7R (Fig. 13) and the MLS data sets (Fig. 14). All remaining results can be found in Fig. S10 in the Supplement. No results are available for the ILAS-II,

MIPAS V5H and SMILES data sets. These cover a too short time period for a drift analysis according to our criteria defined in Sect. 3.4. A summary figure for given data set shows all drifts that are statistically significant at the $2\sigma$ uncertainty level relative to this data set. For better distinction between the data sets results are only plotted at every second altitude, i.e. with a sampling of $\sim 1\,\mathrm{km}$ (see Sect. 3.3 as well as Figs. 4 and 10). The different panels focus on different latitude bands, as indicated in the upper left. For some of the data sets (e.g. GOMOS, HALOE, POAM III, see Supplement) there are occasionally no results for

any of the latitude bands. In these cases there will be information explaining why, in accordance to the list below:

(1) "no comparisons": No comparisons to other data sets could be performed due to missing overlap (at least 20 coincidences, see Sect. 3.3).

(2) "no drift data": Comparisons to other data sets were performed, but yield no drift results. This is because the overlap period is too short or too few data points exist to derive drift estimates.

(3) "no significant results": Drifts were derived, but none of them is statistically significant at the $2\sigma$ uncertainty level.



(4) "significant results only outside the plot range": Statistically significant drifts were derived, but those are outside the plot range from $-3\,\mathrm{ppmv \cdot decade^{-1}}$ to $3\,\mathrm{ppmv \cdot decade^{-1}}$. This is already a large range that covers of the vast majority of reasonable estimates.

In the legend all possible data sets are listed. A colour-coding is used to convey extra information which data sets contribute
results or not (on a global scale):

(1) dark grey: As "no comparisons" in the list above.

(2) light blue: As "no drift data" in the list above.

(3) dark red: As "significant results only outside the plot range" in the list above. This occurs only in a few comparisons (e.g. in the comparison between the GOMOS and SOFIE data sets) and concerns in these cases just a handful of data points. In
the figures shown here in the main manuscript no such case occurs.

(4) black: The comparisons to these data sets yield drifts that are significant at the $2\sigma$ uncertainty level and, thus, these drift results are visible in the given summary figure.

The comparisons to the MIPAS-ESA V7R data set exhibit predominantly negative drifts as visible in Fig. 13. These negative drifts minimise in size typically around $20\,\mathrm{hPa}$ to $10\,\mathrm{hPa}$ (up to $-0.3\,\mathrm{ppmv \cdot decade^{-1}}$ to $-0.4\,\mathrm{ppmv \cdot decade^{-1}}$) and
increase towards lower and higher altitudes. Around $50\,\mathrm{hPa}$ large negative drifts (beyond $-1\,\mathrm{ppmv \cdot decade^{-1}}$) are observed relative to the SMR 489 GHz data set in all latitude bands. Below $100\,\mathrm{hPa}$ drifts of similar size are found more frequently relative to a number of data sets. In the mesosphere again the comparison to SMR 489 GHz data set yield large negative drifts. They often peak in size around $0.5\,\mathrm{hPa}$ (in the polar regions a bit higher up) with values between $-2\,\mathrm{ppmv \cdot decade^{-1}}$ and $-1.2\,\mathrm{ppmv \cdot decade^{-1}}$. At $0.1\,\mathrm{hPa}$ the negative drifts vary between $-1.6\,\mathrm{ppmv \cdot decade^{-1}}$ and $-0.3\,\mathrm{ppmv \cdot decade^{-1}}$,
based on the comparisons to the other MIPAS V5R NOM, MLS and SMR 489 GHz data sets. Notable positive drifts (beyond $1\,\mathrm{ppmv \cdot decade^{-1}}$) are found below $20\,\mathrm{hPa}$ relative to the HIRDLS, GOMOS, MAESTRO data sets at selected latitudes. Smaller positive drifts occur in the comparisons to the MIPAS V5R NOM data sets derived with the Bologna, IMKIAA and Oxford processors as well as some SCIAMACHY data sets. In the latitude bands from 90°S to 60°S and 60°S to 30°S drifts up to $0.5\,\mathrm{ppmv \cdot decade^{-1}}$ are observed in the altitude range from $8\,\mathrm{hPa}$ to $1\,\mathrm{hPa}$ in the comparisons to the ACE-FTS, MLS
and SCIAMACHY lunar (only Antarctic) data sets. Close to $0.1\,\mathrm{hPa}$ often pronounced positive drifts are found relative to the any of MIPAS-Bologna data sets (except in the latitude from 15°S to 15°N and 60°N to 90°N).

The MIPAS V5 data sets are prone to a small drift since the correction coefficients for the non-linearity in the detector response function have changed over time and this is not accounted for in the V5 calibration (Walker and Stiller, in preparation). In the MIPAS V7 calibration a time dependence of these coefficients is considered and, thus, data sets derived from this
calibration are expected to show less drifts. Compared to its predecessor data set, i.e. MIPAS-ESA V5R NOM, the MIPAS-ESA V7R data set exhibits indeed a reduced number of significant drifts. Considering the comparisons to non-MIPAS data sets the number is reduced by 4.6%. If only comparisons to the ACE-FTS v3.5 and MLS data sets are taken into consideration





the reduction amounts to 25%. In contrast to the MIPAS-ESA V7R data set, the MIPAS-ESA V5R NOM data set shows a predominance of positive drifts (see Fig. S10 in the Supplement). This might be a hint that the correction coefficients used in the MIPAS V7 calibration overcompensate the original issue in the V5 calibration.

The comparisons to the MLS data set (Fig. 14) yield both negative and positive drifts with a slight prevalence of the former.
Typically the drift estimates are within $\pm 0.5\,\mathrm{ppmv \cdot decade^{-1}}$. Larger drifts are prominently found in the comparisons to the GOMOS (negative), HIRDLS (primarily positive) and SMR 489 GHz data sets, which highlights rather issues with these data sets than with the MLS data set itself. Positive drifts are consistently found relative to the ACE-FTS data set (around $0.2\,\mathrm{ppmv \cdot decade^{-1}}$). A similar picture is observed in the comparisons to the SCIAMACHY solar occultation data sets. In the lower stratosphere the comparisons to the MIPAS-ESA V5R NOM (roughly up to about 60 hPa), MIPAS-IMKIAA V5R NOM
(up to about 10 hPa) and MIPAS-Oxford V5R NOM (up to about 30 hPa to 20 hPa) exhibit positive drift estimates. In the lower mesosphere the drifts are positive relative to the MIPAS-Bologna, MIPAS-ESA V7R (not in the Antarctic and from 60°S to 30°S) and the MIPAS-IMKIAA V5R NOM (except in the Antarctic) data sets. In addition, the comparisons to the SOFIE data set indicate positive drifts in the Antarctic (with a gap between 9 hPa and 1.5 hPa) and the Arctic lower mesosphere. In contrast, negative drifts are found relative to the MIPAS-Bologna data sets below about 1 hPa, the MIPAS-ESA data sets
(except those positive drifts mentioned before), the V5R MA data sets derived with the IMKIAA and Oxford processors, the MIPAS-Oxford V5R NOM data set above about 30 hPa to 20 hPa as well as the SMR 489 GHz data set.

## 5.4 Method comparison

In this work we present drift results based on coincident observations. In a preceding WAVAS-II work we presented drift estimates among the different data sets based on the comparison of their zonal mean time series (Khosrawi et al., 2018). The
latter approach has the advantage that more data can be used, typically also allowing more comparisons. The disadvantage of the zonal mean time series approach is that it is more prone to sampling errors (in time and space) and does not take into account differences in vertical resolution among the data sets, which, under circumstances, may influence the drift estimates. Here we want to compare the results from these two comparison methods and assess how often the drift estimates differ or not. For that Fig. 15 shows the drift estimates among the different data sets, calculated using the profile-to-profile method, in a matrix form
considering data in the latitude band from 80°S to 70°S at 80 hPa, 10 hPa, 3 hPa and 0.1 hPa (from top to bottom). The drift estimates are based on the difference time series between the data sets given at the x-axis and the data sets given at the y-axis. Data sets are only shown if they yield any result at a given altitude to optimise the available space. The drift estimates are colour-coded. In addition to that, different types of auxiliary information are provided in the result boxes. In the upper left the overlap period of the two data sets is given as well as the number of months the data sets actually overlap. A non-significant
drift (at the $2\sigma$ uncertainty level) is indicated by a slant. For contrast a significant drift is marked by a green frame and the significance level is noted in the lower right corner. As such the figure is directly comparable to Fig. 11 shown in the work of Khosrawi et al. (2018). Both figures exhibit a number of similar patterns. At 0.1 hPa the drift size is the largest among the four altitudes with prominent examples in the comparisons relative to the MIPAS-Bologna V5R MA and SMR 489 GHz data





sets. At $3\,\mathrm{hPa}$ the large drifts relative to SMR 489 GHz are again a common feature among the two drift estimation methods. Smaller drift sizes are observed both at $10\,\mathrm{hPa}$ and $80\,\mathrm{hPa}$ with exceptions attributed to the same data sets.

For a more quantitative comparison, Fig. 16 shows in the same style as Fig. 15 the actual drift differences $\Delta C_{\mathrm{drift}}$ between the two approaches, i.e.:

$$\Delta C_{\mathrm{drift}}(\phi, z) = C_{\mathrm{drift}}^{\mathrm{p2p}}(\phi, z) - C_{\mathrm{drift}}^{\mathrm{zmts}}(\phi, z) \tag{12}$$

Here, $C_{\mathrm{drift}}^{\mathrm{p2p}}$ represents the drifts derived from the profile-to-profile comparisons (this work) and $C_{\mathrm{drift}}^{\mathrm{zmts}}$ the drifts based on the comparisons of the zonal mean time series (Khosrawi et al., 2018). The uncertainty of this difference $\varepsilon_{\mathrm{drift}}$ is given by:

$$\varepsilon_{\mathrm{drift}}(\phi, z) = \sqrt{\varepsilon_{\mathrm{drift}}^{\mathrm{p2p}}(z)^2 + \varepsilon_{\mathrm{drift}}^{\mathrm{zmts}}(\phi, z)^2} \tag{13}$$

where $\varepsilon_{\mathrm{drift}}^{\mathrm{p2p}}$ and $\varepsilon_{\mathrm{drift}}^{\mathrm{zmts}}$ are the drift uncertainties from the two approaches and any covariance among them is neglected. The

characteristic numbers in the result boxes of Fig. 16 correspond to the profile-to-profile comparisons. Differences not statistically significant at the $2\sigma$ uncertainty level are again marked by a slant while for significant differences the significance level is once more provided in the lower right corner. The largest differences between the drift estimates from the two approaches occur at $0.1\,\mathrm{hPa}$ and amount to $0.4\,\mathrm{ppmv} \cdot \mathrm{decade}^{-1}$ on average. Prominent examples are visible in comparisons to the ACE-FTS v3.5, SMR 489 GHz and SOFIE data sets. Also, some comparisons among MIPAS data sets exhibit large differences in

the drift estimates. However, only for the comparison between the MIPAS-ESA V5R MA and MIPAS-Oxford V5R NOM data sets the difference ($2\,\mathrm{ppmv} \cdot \mathrm{decade}^{-1}$) is statistically significant, however just barely (significance level is 2.01). In contrast, at $3\,\mathrm{hPa}$ the drift differences minimise among the four altitudes shown in Fig. 16 (on average $0.15\,\mathrm{ppmv} \cdot \mathrm{decade}^{-1}$). Also, at this altitude there is just one comparison for which the drift estimates obtained from the profile-to-profile and zonal mean time series comparisons differ significantly at the $2\sigma$ uncertainty level, namely for that between the MIPAS-IMKIAA V5R NOM and

the MIPAS-Oxford V5R MA data sets. Again, the significance level is not that large, amounting to 2.1. Larger, non-significant differences are found in comparisons to the ACE-FTS v3.5 and SMR 489 GHz data sets. A similar picture is found at $10\,\mathrm{hPa}$. Comparisons to the SMR 544 GHz data set indicate more prominent differences in the drift estimates from the two approaches. However significant differences are once more restricted to one comparison. In this case it concerns the comparison between the ACE-FTS v3.5 and MLS data sets. The difference is rather small with $0.15\,\mathrm{ppmv} \cdot \mathrm{decade}^{-1}$ and the significance levels is

again just above $2\sigma$, i.e. 2.01. At $80\,\mathrm{hPa}$ the comparisons between the SMR 544 GHz and MIPAS data sets yield significant differences in the drift estimates from the two approaches. They vary between $1\,\mathrm{ppmv} \cdot \mathrm{decade}^{-1}$ and $2.5\,\mathrm{ppmv} \cdot \mathrm{decade}^{-1}$ and have significance levels from 2.2 to 4.6. Overall, in just 2.6% of the comparisons shown in Fig. 16 the drift estimates derived from the profile-to-profile and zonal mean time series comparisons differ significantly at the $2\sigma$ uncertainty level.

The zonal mean time series comparisons presented by Khosrawi et al. (2018) consider in addition the latitude bands from

15°S to 15°N and from 50°N to 60°N. For these latitude bands the figures corresponding to Figs. 15 and 16 are shown in the Supplement (see Figs. S11/12 and S13/14). In summary, for the tropical band the drift differences maximise on average at





80 hPa, close the top of the highly variable tropical tropopause layer. The minimum differences are observed at 3 hPa, as for the latitude band from 80°S to 70°S. In total, in 6.0% of the comparisons the drift estimates from the two approaches differ significantly. For the latitude band from 50°N to 60°N the differences are largest at 0.1 hPa and smallest at 3 hPa. In 3.8% of the comparisons the drift differences are significant at the $2\sigma$ uncertainty level in this latitude band.

For the eight latitude bands primarily considered in this work and altitudes above 100 hPa, overall, in 3.2% of the comparisons the drift estimates derived with the profile-to-profile and zonal mean time series comparisons differ at the $2\sigma$ uncertainty level. The differences in the drift estimates from the two approaches minimise in the altitude range between 5 hPa and 2 hPa (typically, based on the median, between $0.05\,\mathrm{ppmv}\cdot\mathrm{decade}^{-1}$ and $0.1\,\mathrm{ppmv}\cdot\mathrm{decade}^{-1}$). The largest differences are observed towards 100 hPa and 0.1 hPa (typically beyond $0.15\,\mathrm{ppmv}\cdot\mathrm{decade}^{-1}$ median-wise). For individual comparisons the
differences between the drift estimates derived with the profile-to-profile and zonal mean time series comparisons can be significantly larger, as visible in the example shown in Fig. 16.

## 6   Conclusions

In this work biases and drifts among 33 data sets of stratospheric and lower mesospheric water vapour, from 15 different satellite instruments, were assessed using profile-to-profile comparisons. Typically, this observational database exhibits the
largest biases below 70 hPa, both in absolute and relative terms (see Figs. 5 and 6). In contrast, the lowest biases are generally observed between 70 hPa and 5 hPa. Based on the 50% percentile (median) over the different comparison results the typical biases vary between 0.25 ppmv and 0.5 ppmv (5% to 10%) in this altitude region. The smallest biases occur here at low and mid latitudes. Higher up, the biases generally increase accompanied by considerable variations with altitude and latitude band. Typical bias values range from 0.3 ppmv to 1 ppmv (4% to 20%), again based on the 50% percentile. Bias histograms
considering comparison results from all altitudes have the form of a Gaussian (see Fig. 7). For other altitude regions this characteristics is not always given (see Figs. S5 to S7). There is no simple picture which latitude band yields the highest (lowest) occurrences for low (high) biases and vice versa.

In our assessment we find a number of noteworthy data set specific biases (see Figs. 8 and S8). The GOMOS data set shows clear negative biases (exceeding $-1.5$ ppmv) above 50 hPa with pronounced variations among the different latitude bands
considered in this work as well as season. In the lower mesosphere the HALOE data set shows large negative biases of about $-1$ ppmv ($-15\%$). The HIRDLS data set shows distinct positive biases around 100 hPa (roughly around 1 ppmv and 20%). Towards the upper limit of this data set at 10 hPa its biases also show a pronounced latitudinal dependence. For the MAESTRO data set the biases deteriorate at its upper end close to 40 hPa. Already below, the biases in the tropics and subtropics exceed 1 ppmv (20%) on many occasions. Around 1 hPa the biases of SAGE II data set are of the order of 1 ppmv (15%). Both
SMILES data sets show large biases with both signs in the small altitude range where this data data set provides coverage (slightly above 200 hPa to 50 hPa). In the lower stratosphere the SMR 544 GHz data sets show pronounced negative biases, often in excess of $-1.2$ ppmv ($-35\%$). Similarly, the other SMR data set in this assessment, based on water vapour emissions



at 489 GHz, exhibits notable negative biases in the upper stratosphere and lower mesosphere. These biases occasionally exceed $-1\,\mathrm{ppmv}$ $(-15\%)$

For the drift assessment we considered a minimum overlap period of 36 months for the data sets compared with each other. Overall, the observational database shows a wide range of drifts that are statistically significant at the $2\sigma$ uncertainty level. In general, the smallest drifts are found between about $30\,\mathrm{hPa}$ to $10\,\mathrm{hPa}$. In this altitude region the drifts typically do not exceed $0.25\,\mathrm{ppmv}\cdot\mathrm{decade}^{-1}$ $(0.40\,\mathrm{ppmv}\cdot\mathrm{decade}^{-1})$ for the aggregated (unaggregated) global comparisons (see Fig. 11). For other latitude bands the maximum drifts are slightly larger. Histograms considering statistically significant results from all altitudes indicate the largest occurrence for drifts between $0.05\,\mathrm{ppmv}\cdot\mathrm{decade}^{-1}$ and $0.3\,\mathrm{ppmv}\cdot\mathrm{decade}^{-1}$.

All data sets that allowed a drift assessment drifted significantly with respect to at least one other data set. While the comparison of two data sets does not immediately indicate which data set is responsible for any observed drift, comparisons to a multitude of data sets have the potential to obtain a clearer picture. In our analysis the data sets with the most prominent drifts are the GOMOS (primarily negative), HIRDLS (negative), MAESTRO (negative), SMR 544 GHz and SMR 489 GHz (positive) data sets. For the MIPAS V5 data sets a small drift has been expected (Walker and Stiller, in preparation) and is also detected. This can be explained with the calibration of these data sets, which does not account for any time dependence of the correction coefficients for the non-linearity in the detector response. These data sets show primarily positive drifts in the stratosphere, the only exception is the IMKAA V5R NOM data set which exhibits mostly negative drifts. The V7 calibration of the MIPAS data implements a time dependence of the correction coefficients. For the MIPAS-ESA V7R data set the number of significant drifts is indeed reduced compared to the MIPAS-ESA V5R NOM, which is the direct predecessor. The reduction is nearly 5% for the comparisons to non-MIPAS data sets and 25% for the comparisons to the ACE-FTS v3.5 and MLS data sets. The majority of drift estimates for the MIPAS-ESA V7R data set are however negative, in contrast to the predecessor data set. This might suggest that the new correction coefficients implemented in the MIPAS v7 calibration overcompensate the original drift issue. In general, the stratospheric drift estimates for the MIPAS data sets are roughly within $\pm0.5\,\mathrm{ppmv}\cdot\mathrm{decade}^{-1}$. A similar range can be also reported the for the ACE-FTS, MLS, SCIAMACHY and SOFIE data sets. For the MLS data set almost an equivalence of positive and negative drifts is observed. The drift estimates for the data sets from the other three instruments are primarily negative. For the HALOE, POAM III, SAGE II, SAGE III data sets only a limited drift assessment is possible. This is because there are only a few data sets which have sufficiently long overlap periods for a drift estimation. Among them are the SMR data sets that have their own issues. Beyond that there is not always consistency in the results which would provide more certainty towards any potential problem for these four data sets.

Finally, we compared our drift estimates to those derived from comparisons of zonal mean time series as presented by Khosrawi et al. (2018). Our profile-to-profile comparisons have the advantage that they minimise sampling errors among the data sets to be compared. Also, differences in the vertical resolution of the data sets can be taken into account. Comparisons of zonal mean time series typically allow more data to be included, potentially yielding more comparisons. For the three latitude bands ($80°\mathrm{S} - 70°\mathrm{S}$, $15°\mathrm{S} - 15°\mathrm{N}$ and $50°\mathrm{N} - 60°\mathrm{N}$) and four altitudes ($80\,\mathrm{hPa}$, $10\,\mathrm{hPa}$, $3\,\mathrm{hPa}$ and $0.1\,\mathrm{hPa}$) considered in the work of Khosrawi et al. (2018) we found that only in 2.6% to 6.0% of the comparisons the drift estimates derived from the profile-to-profile and zonal mean time series comparisons differed significantly at the $2\sigma$ uncertainty level. For the eight latitude




bands primarily considered in this work and for altitudes above $100\,\mathrm{hPa}$ statistically significant drift differences occurred in 3.2% of the comparisons. Hence, there is largely no need for a specific approach to derive the drift estimates.

In our work consider 13 MIPAS data sets, out 33 data sets in total. Even though, they are based on different measurement modes and processors with different retrieval choices, they exhibit some relative similarity with respect to the remaining data

sets. This behaviour has obvious effects on our results and conclusions. Therefore, the general results and the bias summaries of the individual data sets presented in this work consider an aggregation of the results obtained from MIPAS data sets (see Sect. 3.5). With this aggregation the bias percentiles typically exhibit larger values and a smaller occurrence rate for the smallest biases is observed. In addition, the results vary stronger among the different seasons and latitude bands.

Overall, we find that many data sets are useful for scientific analyses, either considering the observational data themselves

or in connection with modelling results. For scientific studies where the absolute amount of water vapour is the key the data sets listed above with bias issues should be used with caution. Likewise, those data sets mentioned with drift issues should be treated with care if variability beyond 36 months is of interest. Combining the bias and drift characteristics found in this work the altitude range between $50\,\mathrm{hPa}$ and $5\,\mathrm{hPa}$ shows the least number of issues in the observational database, making it most optimal place for scientific analyses regarding stratospheric and lower mesospheric water vapour.

*Data availability.*  Data are available upon request.

*Competing interests.*  The authors declare that they have no conflict of interest.

*Acknowledgements.*  The Atmospheric Chemistry Experiment (ACE), also known as SCISAT, is a Canadian-led mission mainly supported by the Canadian Space Agency and the Natural Sciences and Engineering Research Council of Canada. We appreciate the HALOE Science Team and the many members of the HALOE project for producing and characterising the high quality HALOE data set. We would like to

thank the European Space Agency for making the MIPAS level-1b data set available. MLS data were obtained from the NASA Goddard Earth Sciences and Information Center. Work at the Jet Propulsion Laboratory, California Institute of Technology, was done under contract with the National Aeronautics and Space Administration. SCIAMACHY spectral data have been provided by ESA. M. García-Comas was financially supported by the MINECO under its "Ramon y Cajal" subprogramme, project ESP2014-54362-P and EC FEDER funds. S. Lossow was funded by the "Stratospheric Change and its Role for Climate Prediction" (SHARP) under contract STI 210/9-2. Thanks to Hauke Schmidt

for providing the HAMMONIA data used for the convolution of higher vertically resolved data sets. We want to express our gratitude to SPARC and WCRP (World Climate Research Programme) for their guidance, sponsorship and support of the WAVAS-II programme. The article processing charges for this open-access publication were covered by a Research Centre of the Helmholtz Association.





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



**Table 1.** Overview over the water vapour data sets from satellites used in this study.

| Instrument | Data set | Label | Number |
|---|---|---|---|
| ACE-FTS | v2.2 | ACE-FTS v2.2 | 1 |
| | v3.5 | ACE-FTS v3.5 | 2 |
| GOMOS | LATMOS v6 | GOMOS | 3 |
| HALOE | v19 | HALOE | 4 |
| HIRDLS | v7 | HIRDLS | 5 |
| ILAS-II | v3/3.01 | ILAS-II | 6 |
| MAESTRO | Research | MAESTRO | 7 |
| MIPAS | Bologna V5H v2.3 NOM | MIPAS-Bologna V5H | 8 |
| | Bologna V5R v2.3 NOM | MIPAS-Bologna V5R NOM | 9 |
| | Bologna V5R v2.3 MA | MIPAS-Bologna V5R MA | 10 |
| | ESA V5H v6 NOM | MIPAS-ESA V5H | 11 |
| | ESA V5R v6 NOM | MIPAS-ESA V5R NOM | 12 |
| | ESA V5R v6 MA | MIPAS-ESA V5R MA | 13 |
| | ESA V7R v7 NOM | MIPAS-ESA V7R | 14 |
| | IMKIAA V5H v20 NOM | MIPAS-IMKIAA V5H | 15 |
| | IMKIAA V5R v220/221 NOM | MIPAS-IMKIAA V5R NOM | 16 |
| | IMKIAA V5R v522 MA | MIPAS-IMKIAA V5R MA | 17 |
| | Oxford V5H v1.30 NOM | MIPAS-Oxford V5H | 18 |
| | Oxford V5R v1.30 NOM | MIPAS-Oxford V5R NOM | 19 |
| | Oxford V5R v1.30 MA | MIPAS-Oxford V5R MA | 20 |
| MLS | v4.2 | MLS | 21 |
| POAM III | v4 | POAM III | 22 |
| SAGE II | v7.00 | SAGE II | 23 |
| SAGE III | Solar occultation v4 | SAGE III | 24 |
| SCIAMACHY | Limb v3.01 | SCIAMACHY limb | 25 |
| | Lunar occultation v1.0 | SCIAMACHY lunar | 26 |
| | Solar occultation - OEM v1.0 | SCIAMACHY solar OEM | 27 |
| | Solar occultation - Onion peeling v4.2.1 | SCIAMACHY solar Onion | 28 |
| SMILES | NICT v2.9.2 band A | SMILES-NICT band A | 29 |
| | NICT v2.9.2 band B | SMILES-NICT band B | 30 |
| SMR | v2.0 544 GHz | SMR 544 GHz | 31 |
| | v2.1 489 GHz | SMR 489 GHz | 32 |
| SOFIE | v1.3 | SOFIE | 33 |



**Table 2.** The convolution classes. Based on these differences in the vertical resolution were considered in the comparisons of the data sets. The first four classes consider resolution differences around the hygropause while class V addresses differences at the stratopause and lower mesosphere. Data sets marked by a asterisk have a limited coverage of the hygropause. The consideration of differences in the vertical resolution in comparisons to these data sets may be not necessary, but has been performed just in case.

| class I data sets | class II data sets | class III data sets | class IV data sets | class V data sets |
|---|---|---|---|---|
| $dz \leq 1.6\,km$ | $1.6\,km < dz \leq 3.0\,km$ | $3.0\,km < dz \leq 4.5\,km$ | $dz > 4.5\,km$ | $dz > 6.0\,km$ (above 1 hPa) |
| HIRDLS | GOMOS | ACE-FTS v2.2 | SCIAMACHY limb | MIPAS-Bologna V5H |
| ILAS-II | HALOE | ACE-FTS v3.5 | SCIAMACHY lunar* | MIPAS-Bologna V5R NOM |
| MAESTRO | MIPAS-ESA V5R NOM | MIPAS-Bologna V5H | SMILES-NICT band A | MIPAS-Bologna V5R MA |
| POAM III | MIPAS-ESA V5R MA* | MIPAS-Bologna V5R NOM | SMILES-NICT band B | MIPAS-ESA V5H |
| SAGE II | MLS | MIPAS-Bologna V5R MA* | SMR 544 GHz | MIPAS-ESA V7R |
| SAGE III | | MIPAS-ESA V5H | SMR 489 GHz* | MIPAS-IMKIAA V5H |
| SOFIE* | | MIPAS-ESA V7R | | MIPAS-IMKIAA V5R NOM |
| | | MIPAS-IMKIAA V5H | | MIPAS-Oxford V5H |
| | | MIPAS-IMKIAA V5R NOM | | MIPAS-Oxford V5R NOM |
| | | MIPAS-IMKIAA V5R MA* | | SCIAMACHY solar OEM |
| | | MIPAS-Oxford V5H | | |
| | | MIPAS-Oxford V5R NOM | | |
| | | MIPAS-Oxford V5R MA* | | |
| | | SCIAMACHY solar OEM | | |
| | | SCIAMACHY solar Onion | | |



Table 3: Sources and characteristics of the convolution data used in the comparisons.

| Data set | Source of convolution data | log space | vmr or density | zero a priori |
|---|---|---|---|---|
| ACE-FTS v2.2 | set of generated kernels and a priori data vertical resolution of 3.5 km | no | vmr | yes |
| ACE-FTS v3.5 | set of generated kernels and a priori data vertical resolution of 3.5 km | no | vmr | yes |
| GOMOS-LATMOS | set of generated kernels and a priori data vertical resolution: z $\leq$ 20 km: 2 km, z $\geq$ 30 km: 4 km | no | vmr | yes |
| HALOE | set of generated kernels and a priori data vertical resolution of 2.5 km | no | vmr | yes |
| MIPAS-Bologna V5H | complete set of original kernels and a priori data | no | vmr | no |
| MIPAS-Bologna V5R NOM | complete set of original kernels and a priori data | no | vmr | no |
| MIPAS-Bologna V5R MA | complete set of original kernels and a priori data | no | vmr | no |
| MIPAS-ESA V5H | complete set of original kernels and a priori data | no | vmr | yes |
| MIPAS-ESA V5R NOM | complete set of original kernels and a priori data | no | vmr | yes |
| MIPAS-ESA V5R MA | complete set of original kernels and a priori data | no | vmr | yes |
| MIPAS-ESA V7R | complete set of original kernels and a priori data | no | vmr | yes |
| MIPAS-IMKIAA V5H | complete set of original kernels and a priori data | yes | vmr | yes |
| MIPAS-IMKIAA V5R NOM | complete set of original kernels and a priori data | yes | vmr | yes |
| MIPAS-IMKIAA V5R MA | complete set of original kernels and a priori data | yes | vmr | no |
| MIPAS-Oxford V5H | set of generated kernels and a priori data vertical resolution: p$\geq$ 425 hPa: 2 km, p = 200 hPa: 4 km, p = 3 hPa: 4 km, p = 0.1 hPa: 10 km, p $\leq$ 0.01 hPa: 15 km | no | vmr | yes |
| MIPAS-Oxford V5R NOM | set of characteristic kernels and corresponding a priori data one kernel per 3 months and 5 latitude bands (90°S – 60°S, 60°S – 20°S, 20°S – 20°N, 20°N – 60°N, 60°N – 90°N) | yes | vmr | no |
| MIPAS-Oxford V5R MA | set of generated kernels and a priori data vertical resolution: p$\geq$ 250 hPa: 3 km, p = 100 hPa: 4 km, p = 1 hPa: 4 km, p = 0.1 hPa: 5 km, p $\leq$ 0.01 hPa: 6 km | no | vmr | yes |
| MLS | one characteristic kernel, complete set of a priori data | yes | vmr | no |





| | | | | |
|---|---|---|---|---|
| SCIAMACHY limb | complete set of original kernels and a priori data | yes | density | no |
| SCIAMACHY lunar | one characteristic kernel, complete set of a priori data | yes | density | no |
| SCIAMACHY solar OEM | one characteristic kernel, complete set of a priori data | yes | density | no |
| SCIAMACHY solar Onion | set of generated kernels and a priori data<br>vertical resolution of 4.1 km | no | vmr | yes |
| SMILES-NICT band A | set of generated kernels and a priori data<br>vertical resolution varies, given in the data files | no | vmr | yes |
| SMILES-NICT band B | set of generated kernels and a priori data<br>vertical resolution varies, given in the data files | no | vmr | yes |
| SMR 544 GHz | set of characteristic kernels and corresponding a priori data<br>one kernel per month, 20°latitude band and<br>1 km tropopause height interval | yes | vmr | no |
| SMR 489 GHz | set of characteristic kernels and corresponding a priori data<br>one kernel per month and 20°latitude band | no | vmr | no |





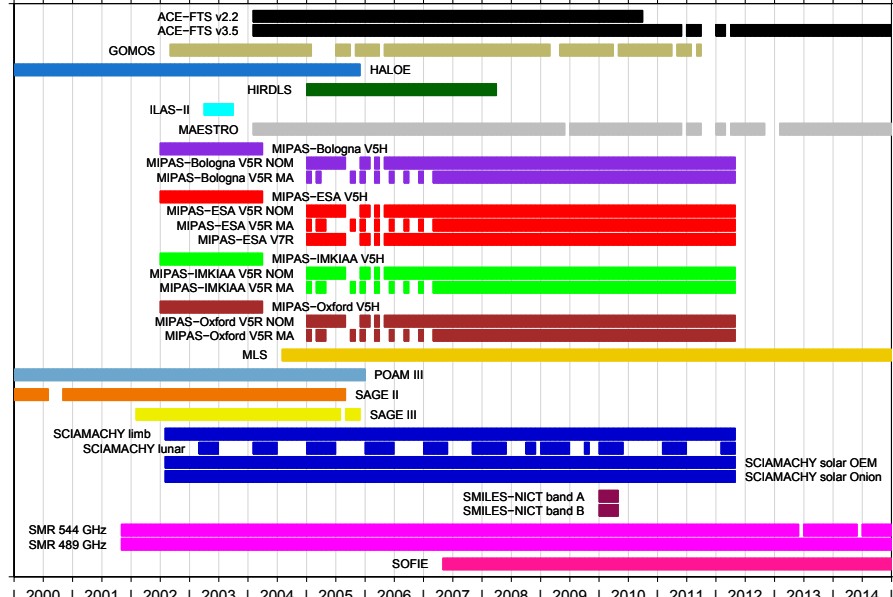

**Figure 1.** The data sets that were included in the comparisons and their corresponding time coverage on a monthly basis. Only data obtained since 2000 are considered.



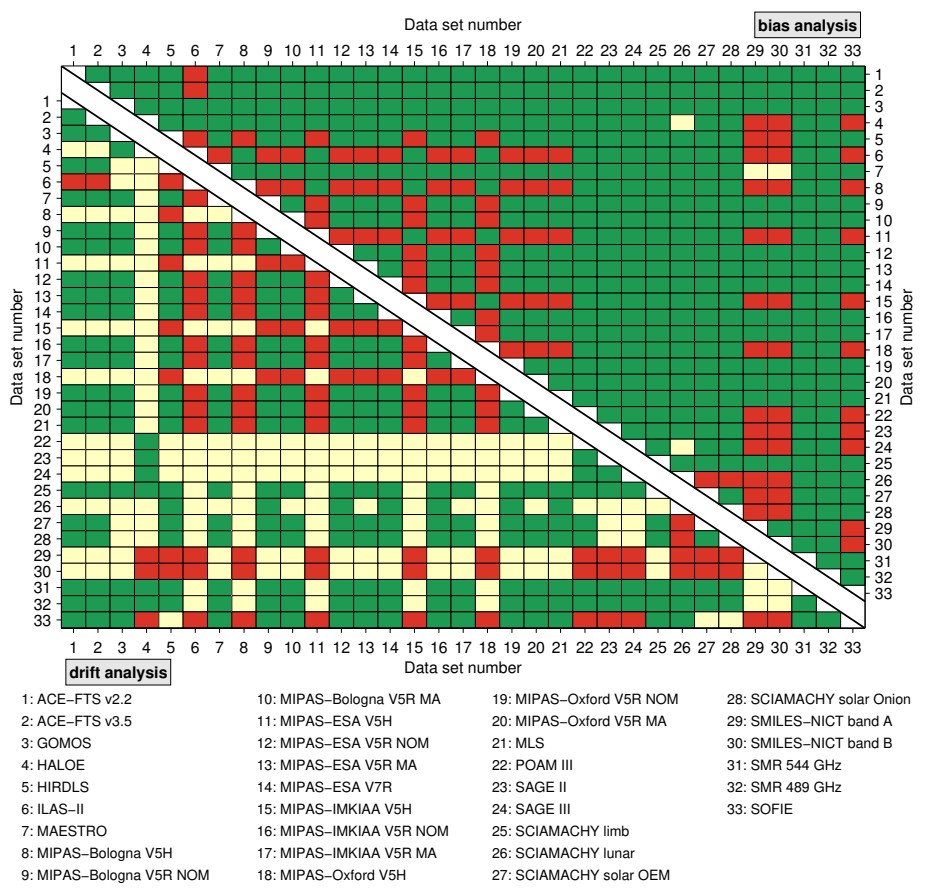

**Figure 2.** Overview which data sets where compared with each other in terms of biases (upper triangle) and drifts (lower triangle). Green means that comparisons were performed while red indicates that this was not the case. Yellow means that comparisons were performed but the results were not considered any further since they did not meet the minimum criteria we defined in Sect. 3. For the bias comparisons this concerns the minimum number of coincidences (i.e. 20, see Sect. 3.3), while for the drift comparisons this involves the minimum overlap period (i.e. 36 months, see Sect. 3.4).





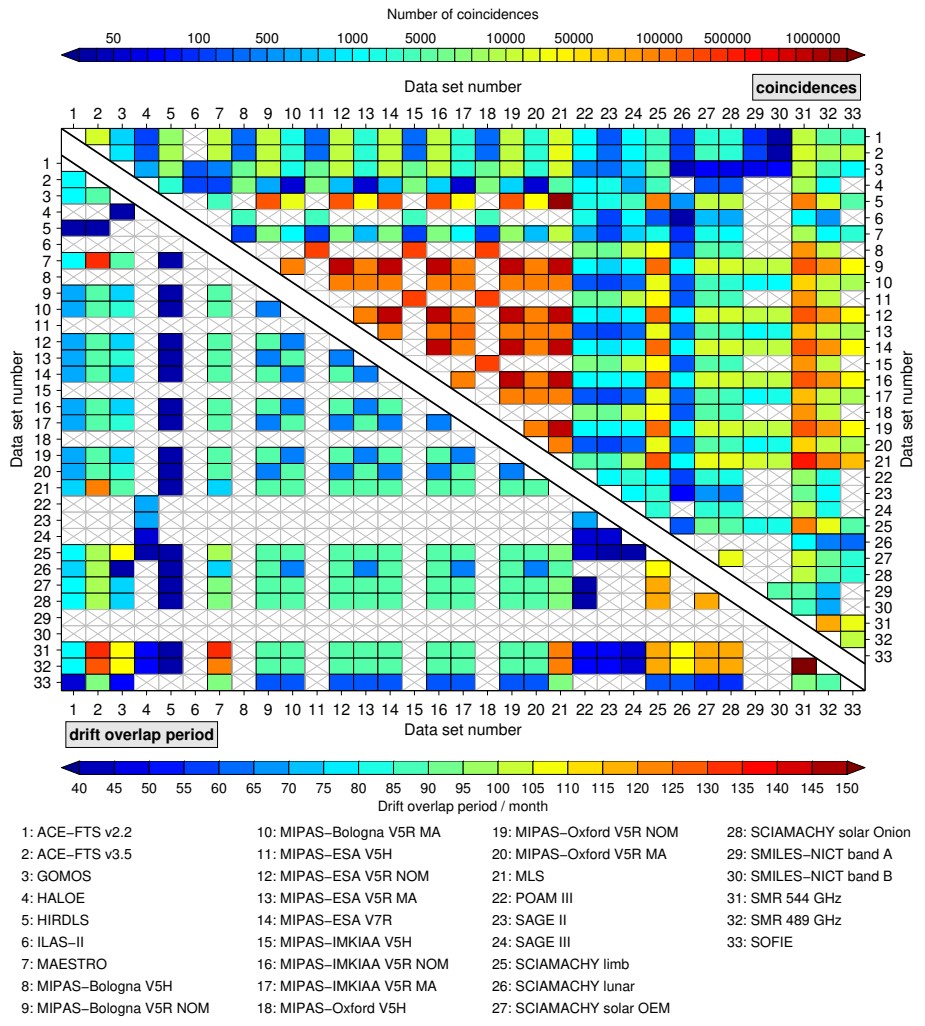

**Figure 3.** Overview on the number of coincidences (upper triangle) and the drift overlap period (lower triangle) between the compared data sets. All numbers consider the comparisons that take into account coincidences during all seasons and at all latitudes. White boxes with grey crosses indicate that no comparison results are available (either yellow or red in Fig. 2).







**Figure 4.** Biases of the SCIAMACHY solar OEM data set in absolute (upper panel) and relative terms (lower panel). These example results are based on coincident observations during all seasons and at all latitudes. The left panel shows the mean biases to the individual data sets, as listed in the legend. In addition, the legend provides information on the temporal and spatial coverage of the individual comparisons. The right panels provides a summary of the bias results shown in the left panels. The red profile is based on the median over all comparisons, while the blue profile considers the aggregation of MIPAS results as described in Sect. 3.5. The latter profile is used in Sect 4.3 and in the Supplement to summarise the bias results for the individual data sets. For better visibility only results at every second altitude are plotted (see Sect. 3.3).





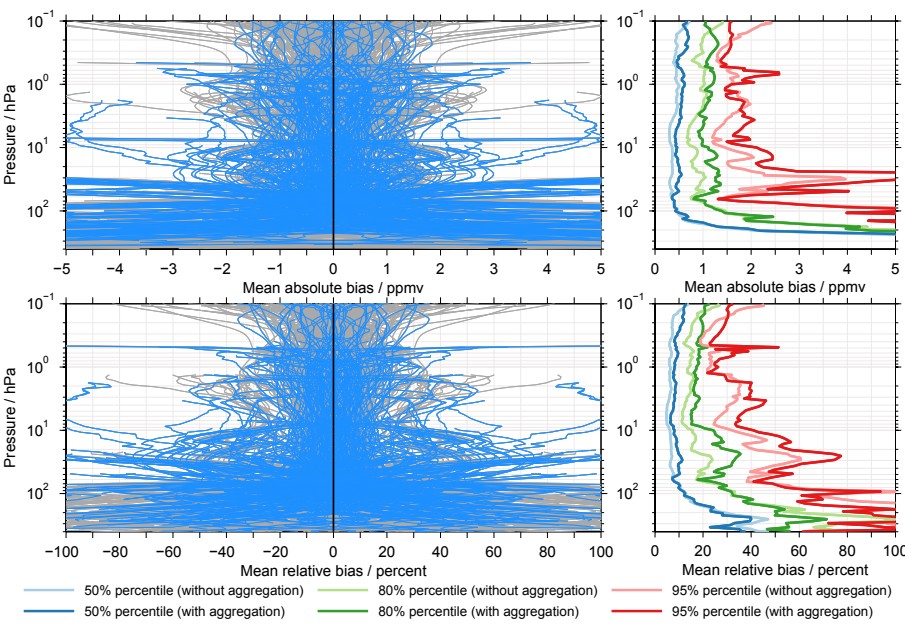

**Figure 5.** Left panel: Bias results for the full matrix of comparisons considering coincidences during all seasons and at all latitudes. The upper panel shows the absolute biases, the lower panel the relative biases. The grey profiles do not consider the aggregation of the MIPAS results while the light blue profiles do. Right panel: The 50% (median), 80% and 95% percentiles derived from the positive part of the biases shown in the left panels, with and without the aggregation of the MIPAS results.



**Figure 6.** The 50% percentile (median) of the biases from all available comparisons for different times and latitude bands, considering the aggregation of the MIPAS results. The left column considers the absolute biases, the right column the relative biases. The different rows focus on individual seasons or their combination. The results for the different latitude bands are colour-coded. On the right-hand side of the individual panels the number of comparisons contributing to the results are indicated. Here, the comparisons to the different MIPAS data sets are counted individually and not combined into a single comparison.



**Figure 7.** Histograms of the absolute (left column) and relative biases (right column) considering results from the entire altitude range. As in the previous figure the different rows consider different seasons while the different latitude bands are colour-coded in the individual panels. Also, these histograms take into consideration the aggregation of the MIPAS results. The increase at the right end of the panes comes from the integration over all biases larger than 3 ppmv and 50%, respectively.





**Figure 8.** A bias summary for all data sets. This summary is based on the comparisons that considers coincidences during all seasons and at all latitudes and takes into account the aggregation of the MIPAS results (see Sect. 3.5). Absolute biases are shown in the left column, relative biases in the right column. For the sake of better visibility the results have been split among three rows. The separation is also reflected by the legend columns. In the legend also the number of comparisons contributing to the summary for the individual data sets is indicated.





**Figure 9.** Bias summary for the ACE-FTS v3.5 data set. The left column considers the absolute biases and the right column the relative biases. As in Fig. 6 and 7 the different rows focus on the results for different seasons or their combination. In the individual panels on the right the number of comparisons contributing to the summary are indicated. As described in Sect. 3.5 the number of comparisons to the different MIPAS data sets are counted individually, even though these results are aggregated.





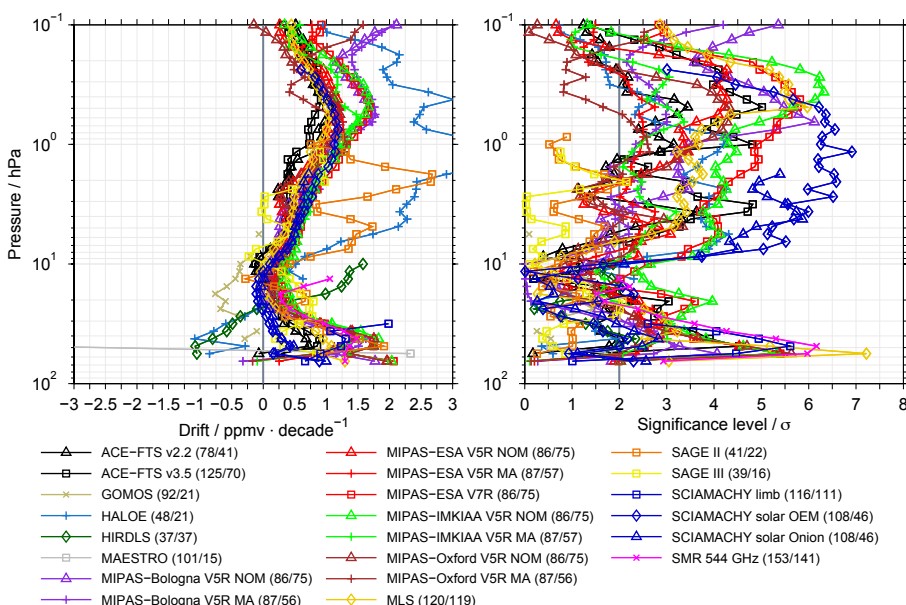

**Figure 10.** The left panel shows the drift of the SMR 489 GHz data set relative to other data sets. In the right panel the corresponding significance level of the drift estimates are shown. This example considers the latitude band between 30°N and 60°N. In the legend the first number indicates the maximum overlap period in terms of months (over all altitudes) of the two data sets compared, i.e. the time between the first and the last month sufficient coincidences were found between the two data sets. The second number indicates during how many months both data sets actually yield sufficient coincidences, again represented by the maximum over all altitudes. As in Fig. 4 only results at every second altitude are plotted for better visibility.




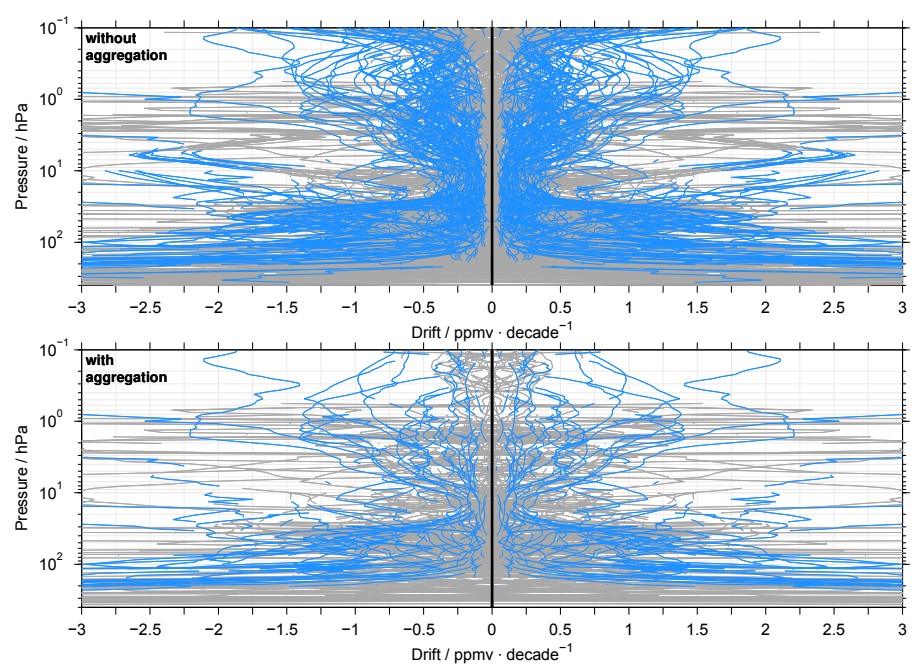

**Figure 11.** Drift results for the full matrix of comparisons considering coincident observations during all seasons and at all latitudes. Drifts that are statistically significant at the $2\sigma$ uncertainty level are marked in light blue. The upper panel shows the picture without the aggregation of the MIPAS results while the lower panel considers this aggregation.



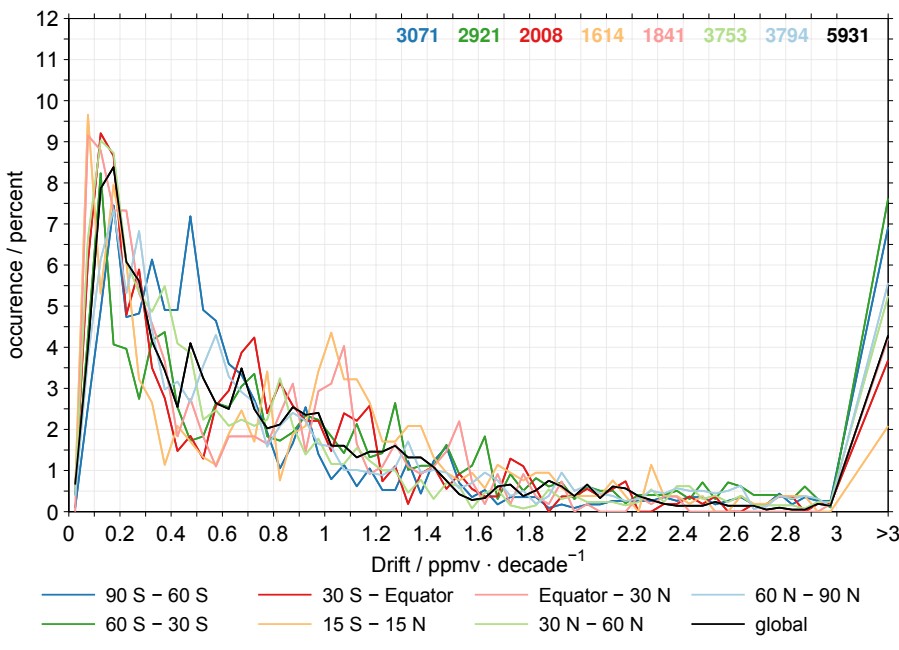

**Figure 12.** Drift histograms using estimates from all altitudes and taking into account the aggregation of the MIPAS results. Only positive drifts and those that are statistically significant at the $2\sigma$ uncertainty level are considered in the calculation. The results for the different latitude bands are given by the different colours. In the upper right corner the number of available data points is indicated, counting comparisons to different MIPAS data sets individually.





**Figure 13.** Drift summary for the MIPAS-ESA v7 data set. The summary shows only drifts that are statistically significant at the $2\sigma$ uncertainty level and only results at every second altitude are plotted (as in Figs. 4 and 10). The different panels consider the results for different latitude bands.The legend lists all possible data sets. Different colours convey additional information which results the different data sets contribute, as described in Sect. 5.3.

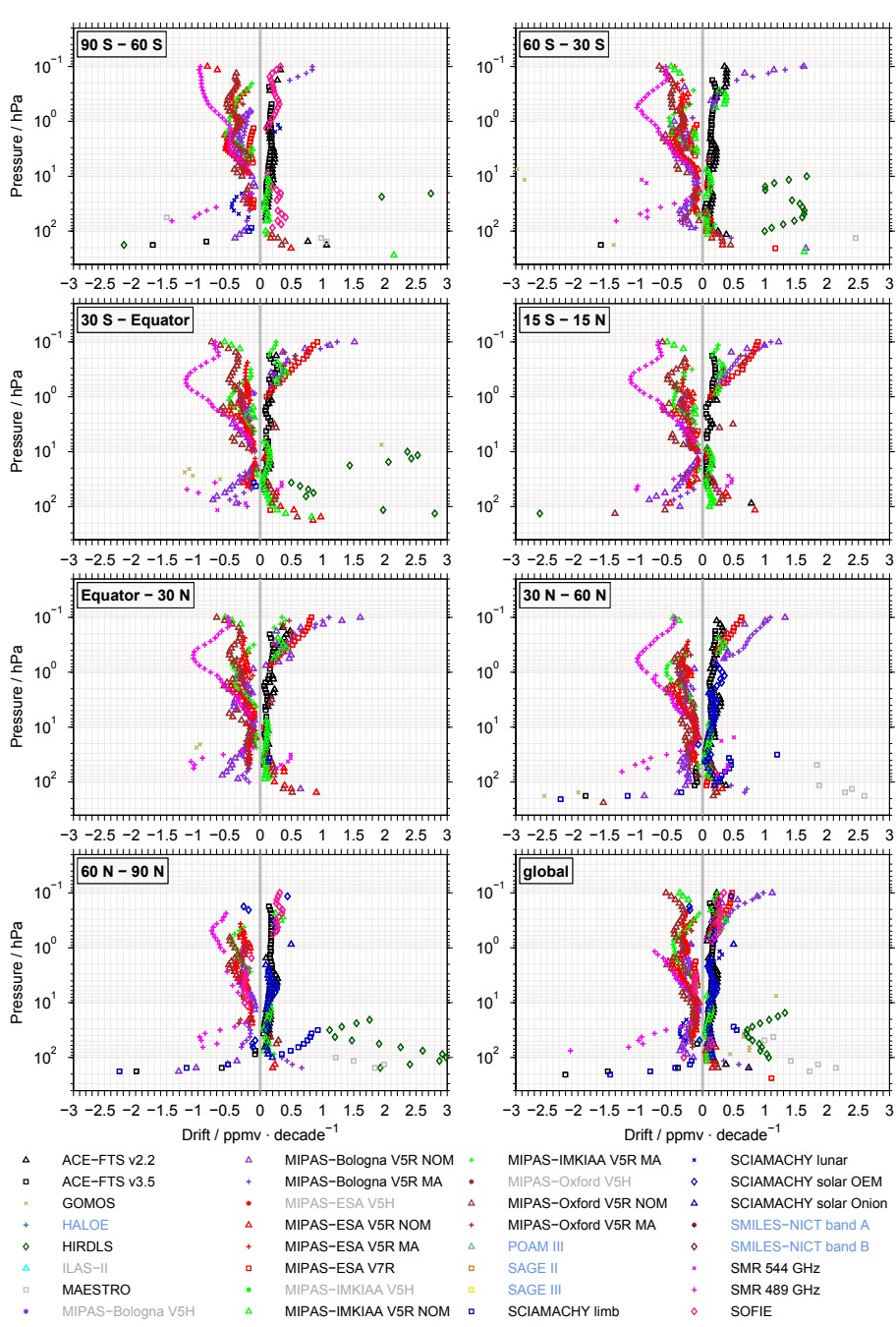

**Figure 14.** As Fig. 13 but here considering the MLS data set.







**Figure 15.** Drifts between the different data sets in the latitude band between $80°$S and $70°$S at $80\,\text{hPa}$, $10\,\text{hPa}$, $3\,\text{hPa}$ and $0.1\,\text{hPa}$. The drift estimates are based on the difference time series between the data sets given at the x-axis and the data sets given at the y-axis. Data sets are only shown if they yield any result at a given altitude. Besides the colour-coded drift estimates the result boxes contain additional information. In the upper left the overlap period of the two data sets is given first. The second number indicates how many months the data sets actually overlap. If a drift is not significant at the $2\sigma$ uncertainty level this is marked by a slant. In contrast, if a drift is significant this is marked by a green frame and the significance level is noted in the lower right corner.



**Figure 16.** As Fig. 15, but here the differences between the drift estimates derived from the profile-to-profile comparisons and those obtained from the comparisons of zonal mean time series (see Eq. 12) are shown. The characteristic numbers in the result boxes correspond to the profile-to-profile comparisons. Typically they are not the same for the comparisons of zonal mean time series and, thus, are just displayed for guidance. Differences not statistically significant at the $2\sigma$ uncertainty level (see Eq. 13) are marked by a slant, otherwise the significance level is again indicated in the lower right corner.