# Peer review of "The SPARC water vapour assessment II"

_Atmospheric Measurement Techniques, 2018_

## Referee Comment (RC1) · Anonymous Referee #1 · 7 Dec 2018

General remarks

This is an impressive and important paper and it must have been a huge task to get all the information and data together. However, the huge dimension of the task is reflected in the length of the paper and thus it is very hard to read. It is not only the length that gives a hard reading, it is also the surprisingly difficult interpretation of "relative bias" and the listing of a very large number of single features of curves that might sometimes be simply noise. Thus there is potential to shorten the paper. I will explain that in more detail below.

[Figure]

Otherwise, I recommend the paper for publication after consideration of the following comments.

Major issues

1) The reader easily gets lost in all the details that the paper provides. Although the conclusion section gives a kind of summary, it is still hard to get an overview for a potential user on which data to use for which purpose. The conclusion section, however, is a good start for this. I suggest to either make lists of bullet points in the last section where information is easier to be found than in the running text, or, even better, a table mentioning advantages and disadvantages for each of the datasets, with respect to both bias and drift issues. If you dare to give recommendations, these would be welcome.

2) Isn't there a lot of averaging in the calculation of biases, for instance? An example where I note this is Fig. 4. Coincidences are found and averaged over all seasons and over the whole latitude band that is available, sometimes $20°$. Doesn't this averaging make the results appear much better than they actually are? A good place to discuss this is in the 2nd paragraph of section 4.1.

3) Equation 8:

I understand your point that none of the two compared datasets should be ennobled as reference. But with your definition of relative bias you get some non-intuitive results that should be mentioned. First, a relative bias undershooting $-100\%$ does not imply that $x_1$ is negative. Instead we have

$$x_1 = (2 + b)/(2 - b)\, x_2,$$

which unfortunately is a quite non-linear function. For instance we get $x_1 = x_2/3$ at a bias of $-100\%$, $x_1 = x_2/2$ at $b = -2/3$, and $x_1 = 3x_2$ at a bias of $+100\%$. In effect, the fair comparison comes at the price of a non-linear and not easily comprehensible relative bias measure. To my impression, all the relative biases given in the paper

easily mislead the reader (and perhaps the authors as well). For instance, in line 8 on page 16 you say "biases ... even exceed ... $\pm 100\%$ in some occasions", which implies that $x_1$ either undershoots $x_2/3$ or exceeds $3x_2$. Is this really what you have intended to say?

4) Page 16, lines 9 to 33: There is a quite detailed description of minor wiggles, local maxima and minima, of the summary bias curves (rhs of Figure 5). Are you sure that these wiggles are physically significant? That is, is there a physical argument that, for instance, the $50\%$ percentile profile minimises at 60 hPa, or isn't this rather incidentally given the sensors that you have in your comparison data base. The non-aggregated line displays different wiggles and it seems to me that you describe noise.

5) Histograms: I doubt that showing and discussing these histograms is of any value. The reader generally wants to know whether a certain data set can be used for an application, that is, whether it has a small bias and low drift in a given altitude (and latitude etc.) region. Perhaps it is important then to have more than one dataset at hand for estimates of uncertainties. But it is not clear to me which purpose the knowledge of these histograms should serve. What, in a concrete situation as sketched, is the gain one has from knowing that the distribution of drift values over 33 data sets peaks in a certain bin? I find this discussion unnecessary and it can be removed without any intellectual loss to the paper.

Furthermore, there are technical issues with the histograms. For instance, the histogram of figure 12 contains about 60 bins which is not ideal. There are several rules for an optimal number of bins, and the result in this case is of the order 15. Consequently there is much noise in the curves and it is not clear whether a single peak as the one described in line 4 on page 22 is real or just noise. A similar comment applies to figure 7, but the curves are much smoother there.

Deleting the histograms and the corresponding text would also have the advantage of getting rid of the unnecessary speculation whether these look Gaussian or not. My

impression is, they do not, but my feeling is also that this an irrelevant point. Whether these distributions are Gaussian or not is not used anywhere in the paper as the basis of an argument.

6) Section 5.4: While it is certainly necessary to compare the methods of drift determination between the current paper and that of Khosrawi et al., I doubt whether it is useful to spend so many lines of text to it and to dwell on so many details. Looking at the corresponding figures, it is evident that the two methods yield essentially consistent results. There are only a handful of exceptions with drift estimates exceeding $2\sigma$. Instead of quoting all the boring numbers (e.g. $\sigma$ is not much larger than 2, namely $2.01$) it would be more useful to think about the possible reasons why the two methods differ for certain data set combinations more than for others. A more fundamental question is whether such differences are generally expected at all and whether a significant deviation is a surprise or not. To my opinion, this section should be rewritten or drastically shortened.

Minor issues

It is a problem that a paper by Walker and Stiller is quite often quoted, which is still in preparation. The reader has no possibility to consult it and does not know when and in which Journal this will eventually become possible. Are there perhaps other "grey" sources of information that may at least partly replace the "in preparation" paper?

Page 3, lines 6-14: it would be nice to have an indication of the relative contribution of each pathway to WV transport into the tropical stratosphere. The sentence "about $3.5$ ppmv to $4.0$ ppmv of water vapour enter the stratosphere" is not understandable. I expect to read something like "per year a total mass of x kilogram of water vapour enters the stratosphere".

Page 6, line 16: I am surprised that mixing ratios between $-20$ and zero are not excluded. Please explain.

Page 7, lines 16 ff: I have only a vague feeling why comparing A to B might give coincidences different from comparing B to A. Perhaps this can be explained a bit clearer. I assume that the "lower half of the comparison matrix" contains just the coincidences of A to B. What then do you mean with "we used those results for the upper half of the matrix"? What does this upper half contain if not the B to A coincidences and in which way can the A to B results be used for the B to A comparisons? And finally, what are the "lower boundaries of the comparisons"?

Page 8, line 1: in which respect is the coverage of the hygropause limited? I don't understand what you mean.

Eq. 1: I wonder why the vectors $x_{high}$ and $x_{apriori}$ have the same number of components although $x_{high}$ refers to high resolution (i.e. more values) while $x_{apriori}$ refers to low resolution (less values).

Page 9, line 25: the following is not an equation, please replace "equation" with "quantity".

Eq. 5: Why do you need the factor $4\ln(2)$ under the $\exp$?

Page 10, beginning of section 3.3: I suggest to write "time period" instead of "time" and "latitude band" instead of "latitude". I was puzzled by assuming that $t$ is a point in time and $\phi$ is a certain latitude and then reading in Eq. 6 that there can be more than one, namely $n_c(t, \phi, z)$ measurements for this point. Later it gets clear, but it is better if it is clear right from the beginning.

Section 3.4: you say $f(...)$ is the regressed bias time series. Before, we had $\bar{b}(...)$ as the bias between two data sets. Unfortunately, it is not very clear to me what the relation is between $f(...)$ and $\bar{b}(...)$. Probably there is one and it would be good to know it.

Page 13, line 16: "comparison results ... are not considered any further". Is this really meant or rather "... are not used for other calculations"? To me, it sounds useless to

compute something that is not considered further.

Page 13, line 27: figures (plural).

Page 14, line 8: "this has not further been pursued".

Page 16, line 11/12: This is not surprising, as this variation is controlled by a smaller and smaller number of outliers.

Page 17, line 11: "within a small altitude range". Actually, this may be a small range in pressure, but certainly quite a large distance in altitude. Please correct.

Fig. 7, right panel: Note that 1% bins are of unequal size in terms of actual differences between $x_1$ and $x_2$. This results from the non-linear character of your bias definition, cf. comment above.

Page 18, line 25: "in the right columns". line 27: "contributing to" and "results are based on the aggregation".

Figure 10: Please provide for the sake of completeness the mathematical definition of $\sigma$. How is it computed for the present application? This should fit after eq. 9.

Figure 11: I wonder why the smaller (blue lines) drifts are statistically significant whereas the larger (grey) ones are not.

Page 24, line 25: is this really from top to bottom? The figure (15) says the opposite. Please check.

Page 27, line 9: please rewrite the sentence. The impression of drifting data sets is surely not what is intended here.

---

## Referee Comment (RC2) · Anonymous Referee #2 · 20 Dec 2018

The paper has an impressive author list and I am sure that the results are worthwhile to publish. But in my mind the pdf-format is not a best way to publish figures including an impenetrable jungle of line plots. You would need a shorter summary in pdf and then an interactive web-page where you could extract detailed results.

---

## Author Comment (AC1) · 17 Feb 2019

**Response to the Comments**
* * *
Color code:

comments of the reviewer
response by the authors
proposed changes in the manuscript
* * *
General comment:

This is an impressive and important paper and it must have been a huge task to get all the information and data together. However, the huge dimension of the task is reflected in the length of the paper and thus it is very hard to read. It is not only the length that gives a hard reading, it is also the surprisingly difficult interpretation of "relative bias" and the listing of a very large number of single features of curves that might sometimes be simply noise. Thus there is potential to shorten the paper. I will explain that in more detail below.

Otherwise, I recommend the paper for publication after consideration of the following comments.

Major comment #1:

The reader easily gets lost in all the details that the paper provides. Although the conclusion section gives a kind of summary, it is still hard to get an overview for a potential user on which data to use for which purpose. The conclusion section, however, is a good start for this. I suggest to either make lists of bullet points in the last section where information is easier to be found than in the running text, or, even better, a table mentioning advantages and disadvantages for each of the datasets, with respect to both bias and drift issues. If you dare to give recommendations, these would be welcome.

Major response #1:

A certain level of detail is clearly intended, primarily for the sake of transparency. Since we are part of an assessment we really like our results to be easily reproduced.

For an overview which data set to use we prefer the summary. In the earlier sections specific results can be only presented for one or two data sets. Which data set is considered varies. This is an attempt to give as many as possible data sets some visibility (across the different papers) and to avoid promotion of a specific data set. In the summary our approach has been to mention data sets if there is something noteworthy about them. If not, they are simply not mentioned. In that sense we clearly prefer a bullet list over a table.

The data set specific results in the summary are provided in a more structured way using bullet lists.

Major comment #2:

Isn't there a lot of averaging in the calculation of biases, for instance? An example where I note this is Fig. 4. Coincidences are found and averaged over all seasons and over the whole latitude band that is available, sometimes 20°. Doesn't this averaging make the results appear much better than they actually are? A good place to discuss this is in the 2nd paragraph of section 4.1.

Major response #2:

In our work we provide mean biases (see Eq. 6) to obtain general results for different combinations of time periods and latitude bands. As described in Sect. 3.3 we require at least 20 coincidences to provide reasonable estimates (other choices could be made). In that sense, yes, there is a lot of averaging involved. But to say that the results appear much better than they actually are is rather odd. Obviously individual comparisons will show smaller biases while other will exhibit larger biases. Individual comparisons are not our interest, but the combination of a bulk of them. Interesting of are course variations among the different time periods and latitude bands as presented in Fig. 9 and Fig. S8.

Major comment #3:

I understand your point that none of the two compared datasets should be ennobled as reference. But with your definition of relative bias you get some non-intuitive results that should be mentioned. First, a relative bias undershooting −100% does not imply that $x_1$ is negative. Instead we have

$$x_1 = \frac{2+b}{2-b} \cdot x_2$$

which unfortunately is a quite non-linear function. For instance we get $x_1 = x_2/3$ at a bias of −100%, $x_1 = x_2/2$ at $b = -2/3$, and $x_1 = 3 \cdot x_2$ at a bias of +100%. In effect, the fair comparison comes at the price of a non-linear and not easily comprehensible relative bias measure. To my impression, all the relative biases given in the paper easily mislead the reader (and perhaps the authors as well). For instance, in line 8 on page 16 you say "biases ... even exceed ... ±100% in some occasions", which implies that $x_1$ either undershoots $x_2/3$ or exceeds $3 \cdot x_2$. Is this really what you have intended to say?

Major response #3:

In fact, this was not our thinking or mind set. If you do an evaluation for single data set then often the data set itself is used as reference. Here we compare a whole matrix of data set combinations. For the relative bias we had multiple options what to be used as reference. For example it could be always the first data set in a comparison, the second data set or the mean among the data sets. We decided to use the mean among the data sets. Besides avoiding any preference of a reference data set this has also practical advantages:

(a) we do not need to know which data set is used as reference in a specific comparison

(b) we do not need to deal with questions regarding the reference when comparing A vs. B or B vs. A (with possible asymmetries)

Given our choice the relative bias is only comprehensible for what it is, i.e. a difference of two data sets about their mean, not data set #1 nor data set #2.

Trying to relating it to data set #1 or data set #2 seems not meaningful nor appropriate. Of course the results for the other reference choices could be calculated. This is however a major effort at this point, let alone integrating the results side by side.

We further motivate our choice and note the non-linearity aspect that was brought up here.

Major comment #4:

Page 16, lines 9 to 33: There is a quite detailed description of minor wiggles, local maxima and minima, of the summary bias curves (rhs of Figure 5). Are you sure that these wiggles are physically significant? That is, is there a physical argument that, for instance, the 50% percentile profile minimises at 60 hPa, or isn't this rather incidentally given the sensors that you have in your comparison data base. The non-aggregated line displays different wiggles and it seems to me that you describe noise.

Major response #4:

It will always be a combination of the data sets involved and physics. The relative importance of these aspects will vary depending on the considered percentile. For the 95% percentile the specific set of data sets considered will certainly have a larger importance than physics.

To provide a more quantitative statement how these percentiles vary we have applied a jackknife approach. Randomly we leave out five data sets and recalculate the percentiles. We repeat this until every data set has been left out at least once. For a given percentile we have then a set of different realisations (typically 25 for the non-aggregated data and a dozen for the aggregated data) from which a standard deviation can calculated. One result is shown in the figure below.

[Figure]

For the 50% percentile the standard deviations are around 0.05 ppmv (1%) without the aggregation of the MIPAS results. With this aggregation the standard deviations are typically twice as large in the stratosphere. Close to 0.1 hPa standard deviations around 0.25 ppmv (4%) are observed. For the 80% percentile the standard deviations amount roughly to 0.1 ppmv (1% - 5%) and 0.2 ppmv (2% - 10%) without and with the aggregation of the MIPAS results, respectively. For the 95% percentile the standard deviations vary typically between 0.05 ppmv and 0.5 ppmv (2% - 10%) when no aggregation of the MIPAS results is considered. If the aggregation is taken account a larger variation is observed with peak values exceeding 1 ppmv (20%). Given the random approach many realisations are possible. The general picture is very similar. The largest variations are visible for 95% percentile.

The results presented here are briefly noted in the manuscript.

Major comment #5:

Histograms: I doubt that showing and discussing these histograms is of any value. The reader generally wants to know whether a certain data set can be used for an application, that is, whether it has a small bias and low drift in a given altitude (and latitude etc.) region. Perhaps it is important then to have more than one dataset at hand for estimates of uncertainties. But it is not clear to me which purpose the knowledge of these histograms should serve. What, in a concrete situation as sketched, is the gain one has from knowing that the distribution of drift values over 33 data sets peaks in a certain bin? I find this discussion unnecessary and it can be removed without any intellectual loss to the paper.

Furthermore, there are technical issues with the histograms. For instance, the histogram of figure 12 contains about 60 bins which is not ideal. There are several rules for an optimal number of bins, and the result in this case is of the order 15. Consequently there is much noise in the curves and it is not clear whether a single peak as the one described in line 4 on page 22 is real or just noise. A similar comment applies to figure 7, but the curves are much smoother there.

Deleting the histograms and the corresponding text would also have the advantage of getting rid of the unnecessary speculation whether these look Gaussian or not. My impression is, they do not, but my feeling is also that this an irrelevant point. Whether these distributions are Gaussian or not is not used anywhere in the paper as the basis of an argument.

Major response #5:

The histograms are presented in a section that focuses on general results. Employing here arguments that emphasise specific data set characteristics does not feel appropriate. We simply had no preconception how such histograms would look like. Also, we wanted to provide another representation of the biases that exist in the observational database, especially comparing the occurrence rate of a given bias as function of latitude band. This is clearly something that is interesting for other multi-data set comparisons, may it be

among satellite data sets, model simulations or a combination of both. We definitely strongly argue against their removal.

In terms of the technical comments we agree the number of bins that used can be reduced. We estimated the number of bins with several rules, i.e. Rice's rule, Scott's rule, Freedman and Diaconis's rule as well as Doane's rule. This was not ultimately helpful. For the bias data (Fig. 7) we found range between 10 to 50 bins depending on rule, number of data points as well the standard deviation or interquartile range of the data. We decided for 0.1 ppmv (0.05 ppmv before) bins for the absolute bias and 2% (1% before) for the relative bias. For the drift data (Fig. 12) the different rules yield a similar picture. In this case we decided for 0.1 ppmv/decade bin (0.05 ppmv/decade before).

There is not so much speculation whether these histograms are Gaussian or not. We fitted a Gaussian function to the histograms, but decided not to show these results to keep number of lines to a reasonable level. Arguably the fits are better at larger biases/drifts than at lower values. We decided to remove this argumentation.

The histograms are recalculated and all related text is adapted. The argumentation on the Gaussian shape of the histograms is removed.

Major comment #6:

Section 5.4: While it is certainly necessary to compare the methods of drift determination between the current paper and that of Khosrawi et al., I doubt whether it is useful to spend so many lines of text to it and to dwell on so many details. Looking at the corresponding figures, it is evident that the two methods yield essentially consistent results. There are only a handful of exceptions with drift estimates exceeding $2\sigma$. Instead of quoting all the boring numbers (e.g. $\sigma$ is not much larger than 2, namely 2.01) it would be more useful to think about the possible reasons why the two methods differ for certain data set combinations more than for others. A more fundamental question is whether such differences are generally expected at all and whether a significant deviation is a surprise or not. To my opinion, this section should be rewritten or drastically shortened.

Major response #6:

Also in this case we had no real preconceptions how similar or how different the drift estimates derived from the two approaches would be. It was probably a surprise that so few statistically significant differences exist. We agree to reduce the amount of detail that has been spent on describing the figures. In addition, we performed more analyses, trying to find any prominent behaviour. The differences among the drift estimates from the two approaches are on average largest for comparisons among sparse data sets (i.e. ACE-FTS, GOMOS, HALOE MAESTRO, POAM, SAGE, SCIAMACHY occultation and SOFIE) and smallest for comparisons among dense data sets (HIRDLS, MIPAS, MLS, SCIAMACHY limb, SMR). At the same time the percentage of significant differences in the drift estimates is smallest for comparisons between sparse data sets and largest for comparisons between dense data sets. Comparisons between dense and sparse data sets yield statistics in the middle. This behaviour is intuitive, but the numbers also support this relation in reality.

The detailed description of Figs. 15 and 16 is shortened. The results from the additional analyses described here shortly will be added.

Minor issue #1:

It is a problem that a paper by Walker and Stiller is quite often quoted, which is still in preparation. The reader has no possibility to consult it and does not know when and in which Journal this will eventually become possible. Are there perhaps other "grey" sources of information that may at least partly replace the "in preparation" paper?

Minor response #1:

Within the WAVAS-II activity is was decided to have such data set overview paper to avoid describing the data set all over again and to have a central reference. Critically, we have to admit that this should have been one of the first papers, but it will be one of the last instead. The following publications (certainly not a complete list) are of relevance:

| | | |
|---|---|---|
| (1) Azam et al. (2012) | (11) Montoux et al. (2009) | |
| (2) Carleer et al. (2008) | (12) Noël et al. (2010) | |
| (3) Dinelli et al. (2010) | (13) Raspollini et al. (2013) | |
| (4) Eriksson et al. (2014) | (14) Rong et al. (2010) | |
| (5) Gille et al. (2013) | (15) Sioris et al. (2016) | |
| (6) Griesfeller et al. (2008) | (16) Taha et al. (2004) | |
| (7) Hegglin et al. (2013) | (17) Thomason et al. (2010) | |
| (8) Kley et al. (2000) | (18) von Clarmann et al. (2009) | |
| (9) Livesey et al. (2015) | (19) Weigel et al. (2016) | |
| (10) Lumpe et al. (2006) | (20) Urban et al. (2007) | |

The full references are provided at the end of this document.

Minor issue #2:

Page 3, lines 6-14: it would be nice to have an indication of the relative contribution of each pathway to WV transport into the tropical stratosphere. The sentence "about 3.5 ppmv to 4.0 ppmv of water vapour enter the stratosphere" is not understandable. I expect to read something like "per year a total mass of x kilogram of water vapour enters the stratosphere".

Minor response #2:

Currently we cannot give exact numbers on the relative importance of the different pathways contributing to the transport of water vapour from the troposphere to the stratosphere. This is an important topic of ongoing research. Slow ascent certainly plays the most important role and the relative importance will vary throughout the year.

The 3.5 ppmv to 4.0 ppmv describe the typical entry mixing ratio of water vapour in the tropical stratosphere (averaged over a year). This is a common notion. Of course we could provide a total mass in kilogramme, however this would quite unusual in this context and especially when it comes to satellite observations.

The text is changed to: "Overall, the stratospheric entry mixing ratios typically amount to 3.5 ppmv to 4.0 ppmv, on an annual average (Kley et al., 2000)."

Minor issue #3:

Page 6, line 16: I am surprised that mixing ratios between −20 and zero are not excluded. Please explain.

Minor response #3:

Negative volume mixing ratios can occur due to noise in the observations which propagates through the retrievals. As such, this is an important information that should not be removed. In addition, screening these negative values can create biases that are not correct.

Minor issue #4:

Page 7, lines 16 ff: I have only a vague feeling why comparing A to B might give coincidences different from comparing B to A. Perhaps this can be explained a bit clearer. I assume that the "lower half of the comparison matrix" contains just the coincidences of A to B. What then do you mean with "we used those results for the upper half of the matrix"? What does this upper half contain if not the B to A coincidences and in which way can the A to B results be used for the B to A comparisons? And finally, what are the "lower boundaries of the comparisons"?

Minor response #4:

The sketch below provides a simple example of different coincidences for comparisons A vs. B and B vs. A. As described in the manuscript the data sets sorted chronologically. For the comparison A vs. B we go through the individual observations of A (here described by the subscripts 1 to 4) and try to find observations of B that fulfil the coincidence criteria. For observation $A_1$ only one coincidence is found, namely $B_1$ (marked by the solid arrow). Next in line is observation $A_2$. In this case three coincident observations of B are found. In this case we chose the observations closest in space as coincidence, which is here $B_3$. For the observations $A_3$ and $A_4$ no coincident observations of B are found. Overall, we find two coincidences for this simple comparison: $A_1$ vs. $B_1$ and $A_2$ vs. $B_3$. When B vs. A is compared the observations $B_1$ and $A_1$ are coincident again. For observation $B_2$ only $A_2$ is found as coincidence. Next is observation $B_3$. In this case also observation $A_2$ fulfils the coincidence criteria. However since $A_2$ was already considered as a coincidence (to $B_2$) it is not further considered (marked by the dotted line). This is what we denote in the manuscript as the unique coincidence approach (i.e. no observation is used twice in a comparison). For observation $B_4$ there is the same situation as for $B_3$. Overall, we also find two coincidences for the comparison B vs. A: $B_1$ vs. $A_1$ and $B_2$ vs. $A_2$. However, they are different than for A vs. B. In general, not only the exact coincidences can be different, but also their absolute number.

[Figure]

Regarding the lower and upper half of the comparison matrix an example is given below considering four data sets A, B, C and D. We have only performed

comparison in the lower half of matrix, here marked in blue. For the upper half, here marked in red, we used the results from the lower half, but of course with the opposite sign.

| x | A | B | C | D |
|---|---|---|---|---|
| A | x | -(A - B) | -(A - C) | -(A - D) |
| B | A - B | x | -(B - C) | -(B - D) |
| C | A - C | B - C | x | -(C - D) |
| D | A - D | B - D | C - D | x |

The term "lower boundaries of the comparisons" is supposed to described the lowermost altitudes where comparisons were possible.

The text is changed to: "Larger deviations were mostly found at the lower altitude limits of the comparisons."

Minor issue #5:

Page 8, line 1: in which respect is the coverage of the hygropause limited? I don't understand what you mean.

Minor response #5:

The data sets in question have all their lower altitude limit (where reasonable retrievals are possible) close to the hygropause. In that sense the observational coverage is meant. Some retrieved profiles will cover the hygropause, others not.

The text is changed to: "In the table columns some data sets have been marked by an asterisk, indicating that these data sets have a limited observational coverage of the hygropause. Some retrieved profiles will include the hygropause, others not. Hence, some comparisons to these data sets may not necessarily need the consideration of differences in the vertical resolution in this altitude range."

Minor issue #6:

Eq. 1: I wonder why the vectors $x_{high}$ and $x_{apriori}$ have the same number of components although $x_{high}$ refers to high resolution (i.e. more values) while $x_{apriori}$ refers to low resolution (less values).

Minor response #6:

The relation that is indicated here does not need to exist. Results with low vertical resolution can be retrieved on a fine grid (corresponding to more values). Prominent examples are the MIPAS data sets retrieved with the IMKIAA processor. The retrieval uses a fixed 1 km grid through most of the stratosphere while the vertical resolution varies between 3 km and 5 km. As described by Eqs. 1 and 2 the convolution is performed on the grid of the lower resolved data set. Regridding the high resolved data set to the grid of the lower resolved data set follows the work of Stiller et al. (2012).

The approach to regrid the high resolved data set to the grid of the lower resolved data is mentioned.

Minor issue #7:

Page 9, line 25: the following is not an equation, please replace "equation" with "quantity".

Minor response #7:

Okay.

The text is changed accordingly.

Minor issue #8:

Eq. 5: Why do you need the factor 4 ln(2) under the exp?

Minor response #8:

The starting point is the basic Gauss function:

$$G(\mathbf{x}, \mu, \sigma) = \exp\left\{ -\frac{[\mathbf{x} - \mu]^2}{2 \cdot \sigma^2} \right\}$$

In our case we have substituted the vector $\mathbf{x}$ with the altitude vector $\mathbf{z}$. The average $\mu$ is used as the specific altitude for which the kernel is created, i.e. $\mathbf{z}(j)$. Furthermore we consider the altitude resolution $\mathbf{dz}(j)$ (i.e. full width at half maximum) at the specific altitude instead of the standard deviation $\sigma$. This gives:

$$G[\mathbf{z}, \mathbf{z}(j), \sigma] = \exp\left\{ -\frac{[\mathbf{z} - \mathbf{z}(j)]^2}{2 \cdot \sigma^2} \right\} \quad \text{with}$$

$$\sigma = \frac{\mathbf{dz}(j)}{2 \cdot \sqrt{2 \cdot ln(2)}} \quad \text{or} \quad \sigma^2 = \frac{\mathbf{dz}(j)^2}{8 \cdot ln(2)} \quad \text{yields}$$

$$G[\mathbf{z}, \mathbf{z}(j), \mathbf{dz}(j)] = \exp\left\{ -\frac{4 \cdot \ln(2) \cdot [\mathbf{z} - \mathbf{z}(j)]^2}{\mathbf{dz}(j)^2} \right\}$$

Minor issue #9:

Page 10, beginning of section 3.3: I suggest to write "time period" instead of "time" and "latitude band" instead of "latitude". I was puzzled by assuming that t is a point in time and $\phi$ is a certain latitude and then reading in Eq. 6 that there can be more than one, namely $n_c(t, \phi, z)$ measurements for this point. Later it gets clear, but it is better if it is clear right from the beginning.

Minor response #9:

We agree.

The text is adapted accordingly.

Minor issue #10:

Section 3.4: you say $f(...)$ is the regressed bias time series. Before, we had $b(...)$ as the bias between two data sets. Unfortunately, it is not very clear to me what the relation is between $f(...)$ and $b(...)$. Probably there is one and it would be good to know it.

Minor response #10:

$f(t, \phi, z)$ and $\overline{b}(t, \phi, z)$ are of course related. $f(t, \phi, z)$ represents the fit of regressed time series of $\overline{b}(t, \phi, z)$, however in this case $t$ describes not a specific season or the entire year but all months which the data sets compared have sufficient overlap (i.e. at least 5 coincidences, see Sect. 3.4). We do not want to change the regression equation as it has been used in different WAVAS publications, avoiding the creation of any unnecessary inconsistency.

The text is changed to: "In the equation $f(t, \phi, z)$ represents the fit of the regressed bias time series $\overline{b}(t, \phi, z)$, however here $t$ describes all months where the data sets that are compared have sufficient overlap (i.e. 5 coincidences, see above)."

Minor issue #11:

Page 13, line 16: "comparison results ... are not considered any further". Is this really meant or rather "... are not used for other calculations"? To me, it sounds useless to compute something that is not considered further.

Minor response #11:

The comparison results among the MIPAS data sets are only used the non-aggregated results. For the aggregated results they are not used, as the aggregation treats the MIPAS data sets as if they were only one data set. Accordingly, the comparisons among them are not relevant.

The text is changed to: "comparison results between different MIPAS data sets are not considered (in the calculation of the aggregated quantities)".

Minor issue #12:

Page 13, line 27: figures (plural).

Minor response #12:

Thanks!

The text is changed accordingly.

Minor issue #13:

Page 14, line 8: "this has not further been pursued".

Minor response #13:

Done.

The text is changed accordingly.

Minor issue #14:

Page 16, line 11/12: This is not surprising, as this variation is controlled by a smaller and smaller number of outliers.

Minor response #14:

We totally agree, it just describes the observation.

Minor issue #15:

Page 17, line 11: "within a small altitude range". Actually, this may be a small range in pressure, but certainly quite a large distance in altitude. Please correct.

Minor response #15:

Yes, this could be misleading.

The phrase is removed.

Minor issue #16:

Fig. 7, right panel: Note that 1% bins are of unequal size in terms of actual differences between $x_1$ and $x_2$. This results from the non-linear character of your bias definition, cf. comment above.

Minor response #16:

Please see our major response #3.

Minor issue #17:

Page 18, line 25: "in the right columns".

Minor response #17:

Okay.

The word "right" is added and "columns" is corrected to "column".

Minor issue #18:

Page 18, line 27: "contributing to" and "results are based on the aggregation".

Okay.

The first part is changed accordingly. The second part is added but just with parentheses.

Minor issue #19:

Figure 10: Please provide for the sake of completeness the mathematical definition of σ. How is it computed for the present application? This should fit after eq. 9.

Minor response #19:

Okay.

The equation is provided at the end of section 3.4.

Minor issue #20:

Figure 11: I wonder why the smaller (blue lines) drifts are statistically significant whereas the larger (grey) ones are not.

Minor response #20:

These cases can actually be attributed to specific data sets. Throughout the stratosphere this concerns comparisons to the GOMOS data set. This data set exhibits a low precision, resulting both in large drift and associated uncertainty estimates. In the lower stratosphere, such behaviour occurs in a significantly increased manner also in comparisons to the HIRDLS, MAESTRO and SMR 544 GHz data sets. All three data sets have their upper vertical limit in this altitude region where they exhibit increased uncertainties.

Minor issue #21:

Page 24, line 25: is this really from top to bottom? The figure (15) says the opposite. Please check.

Minor response #21:

This is a typo. Thank you for spotting!

The text is corrected.

Minor issue #22:

Page 27, line 9: please rewrite the sentence. The impression of drifting data sets is surely not what is intended here.

Minor response #22:

This is certainly an unfortunate expression.

It is removed in the rework of the conclusion as described in the major response #1.

**References**

Azam, F., Bramstedt, K., Rozanov, A., Weigel, K., Bovensmann, H., Stiller, G. P., and Burrows, J. P.: SCIAMACHY lunar occultation water vapor measurements: retrieval and validation results, Atmospheric Measurement Techniques, 5, 2499 – 2513, https://doi.org/10.5194/amt-5-2499-2012, 2012.

Carleer, M. R., Boone, C. D., Walker, K. A., Bernath, P. F., Strong, K., Sica, R. J., Randall, C. E., Vömel, H., Kar, J., Höpfner, M., Milz, M., von Clarmann, T., Kivi, R., Valverde-Canossa, J., Sioris, C. E., Izawa, M. R. M., Dupuy, E., McElroy, C. T., Drummond, J. R., Nowlan, C. R., Zou, J., Nichitiu, F., Lossow, S., Urban, J., Murtagh, D. P., and Dufour, D. G.: Validation of water vapour profiles from the Atmospheric Chemistry Experiment (ACE), Atmospheric Chemistry & Physics Discussions, 8, 4499 – 4559, 2008.

Dinelli, B. M., Arnone, E., Brizzi, G., Carlotti, M., Castelli, E., Magnani, L., Papandrea, E., Prevedelli, M., and Ridolfi, M.: The MIPAS2D database of MIPAS/ENVISAT measurements retrieved with a multi-target 2-dimensional tomographic approach, Atmospheric Measurement Techniques, 3, 355 – 374, 2010.

Eriksson, P., Rydberg, B., Sagawa, H., Johnston, M. S., and Kasai, Y.: Overview and sample applications of SMILES and Odin-SMR retrievals of upper tropospheric humidity and cloud ice mass, Atmospheric Chemistry & Physics, 14, 12613 – 12629, https://doi.org/10.5194/acp-14-12613-2014, 2014.

Gille, J., Grey, L., Cavanaugh, C., Coffey, M., Dean, V., Halvorson, C., Karol, S., Khosravi, R., Kinnison, D., Massie, S., Nardi, B., Rivas, M. B., , Smith, L., Torpy, B., Waterfall, A., and Wright, C.: HIRDLS data description and quality version 7, http://docserver.gesdisc.eosdis.nasa.gov/repository/Mission/HIRDLS/3.3_ Product_Documentation/3.3.5_Product_Quality/HIRDLS-DQD_V7.pdf, Last access: 17 January 2018, 2013.

Griesfeller, A., von Clarmann, T., Griesfeller, J., Höpfner, M., Milz, M., Nakajima, H., Steck, T., Sugita, T., Tanaka, T., and Yokota, T.: Intercomparison of ILAS-II version 1.4 and version 2 target parameters with MIPAS-Envisat measurements, Atmospheric Chemistry & Physics, 8, 825 – 843, 2008.

Hegglin, M. I., Tegtmeier, S., Anderson, J., Froidevaux, L., Fuller, R., Funke, B., Jones, A., Lingenfelser, G., Lumpe, J., Pendlebury, D., Remsberg, E., Rozanov, A., Toohey, M., Urban, J., Clarmann, T., Walker, K. A., Wang, R., and Weigel, K.: SPARC Data Initiative: Comparison of water vapor climatologies from international satellite limb sounders, Journal of Geophysical Research, 118, 11 824, https://doi.org/10.1002/jgrd. 50752, 2013.

Kley, D., Russell, J. M., and Philips, C.: Stratospheric Processes and their Role in Climate (SPARC) - Assessment of upper tropospheric and stratospheric water vapour, SPARC Report 2, WMO/ICSU/IOC World Climate Research Programme, Geneva, 2000.

Livesey, N. J., Read, W. J., Wagner, P. A., Froidevaux, L., Lambert, A., Manney, G. L., Millan, L. F., Pumphrey, H. C., Santee, M. L., Schwartz, M. J., Wang, S., Fuller, R. A., Jarnot, R. F., Knosp, B. W., , and Martinez, E.: Aura/MLS version 4.2x level 2 data quality and description document, http://mls.jpl.nasa.gov/data/ v4-2_data_ quality_document.pdf, Last access: 18 August 2017, 2015.

Lumpe, J., Bevilacqua, R., Randall, C., Nedoluha, G., Hoppel, K., Russell, J. M., Harvey, V. L., Schiller, C., Sen, B., Taha, G., Toon, G., and Vömel, H.: Validation of Polar Ozone and Aerosol Measurement (POAM) III version 4 stratospheric water vapor, Journal of Geophysical Research, 111, 11 301, https://doi.org/ 10.1029/2005JD006763, 2006.

Montoux, N., Hauchecorne, A., Pommereau, J.-P., Lefêvre, F., Durry, G., Jones, R. L., Rozanov, A., Dhomse, S., Burrows, J. P., Morel, B., and Bencherif, H.: Evaluation of balloon and satellite water vapour measurements in the Southern tropical and subtropical UTLS during the HIBISCUS campaign, Atmospheric Chemistry & Physics, 9, 5299 – 5319, 2009.

No¨el, S., Bramstedt, K., Rozanov, A., Bovensmann, H., and Burrows, J. P.: Water vapour profiles from SCIAMACHY solar occultation measurements derived with an onion peeling approach, Atmospheric Measurement Techniques, 3, 523 – 535, 2010.

Raspollini, P., Carli, B., Carlotti, M., Ceccherini, S., Dehn, A., Dinelli, B. M., Dudhia, A., Flaud, J.-M., López-Puertas, M., Niro, F., Remedios, J. J., Ridolfi, M., Sembhi, H., Sgheri, L., and von Clarmann, T.: Ten years of MIPAS measurements with ESA Level 2 processor V6 - Part 1: Retrieval algorithm and diagnostics of the products, Atmospheric Measurement Techniques, 6, 2419 – 2439, https://doi.org/10.5194/amt-6-2419-2013, 2013.

Rong, P., Russell, J. M., Gordley, L. L., Hervig, M. E., Deaver, L., Bernath, P. F., and Walker, K. A.: Validation of v1.022 mesospheric water vapor observed by the SOFIE instrument on the AIM Satellite, Journal of Geophysical Research, 115, D16 209, https://doi.org/10.1029/2010JD013852, 2010.

Sioris, C. E., Zou, J., Plummer, D. A., Boone, C. D., McElroy, C. T., Sheese, P. E., Moeini, O., and Bernath, P. F.: Upper tropospheric water vapour variability at high latitudes - Part 1: Influence of the annular modes, Atmospheric Chemistry & Physics, 16, 3265 – 3278, https://doi.org/10.5194/acp-16-3265-2016, 2016.

Stiller, G. P., Kiefer, M., Eckert, E., von Clarmann, T., Kellmann, S., García-Comas, M., Funke, B., Leblanc, T., Fetzer, E., Froidevaux, L., Gomez, M., Hall, E., Hurst, D., Jordan, A., Kämpfer, N., Lambert, A., McDermid, I. S., McGee, T., Miloshevich, L., Nedoluha, G., Read, W., Schneider, M., Schwartz, M., Straub, C., Toon, G., Twigg, L. W., Walker, K., and Whiteman, D. N.: Validation of MIPAS IMK/IAA temperature, water vapor, and ozone profiles with MOHAVE-2009 campaign measurements, Atmospheric Measurement Techniques, 5, 289 – 320, https://doi.org/ 10.5194/amt-5-289-2012, 2012.

Taha, G., Thomason, L. W., and Burton, S. P.: Comparison of Stratospheric Aerosol and Gas Experiment (SAGE) II version 6.2 water vapor with balloon-borne and space- based instruments, Journal of Geophysical Research, 109, D18313, https://doi.org/ 10.1029/2004JD004859, 2004.

Thomason, L. W., Moore, J. R., Pitts, M. C., Zawodny, J. M., and Chiou, E. W.: An evaluation of the SAGE III version 4 aerosol extinction coefficient and water vapor data products, Atmospheric Chemistry & Physics, 10, 2159 – 2173, 2010.

Urban, J., Lautie, N., Murtagh, D. P., Eriksson, P., Kasai, Y., Lossow, S., Dupuy, E., de La Nöe, J., Frisk, U., Olberg, M., Le Flochmöen, E., and Ricaud, P.: Global observations of middle atmospheric water vapour by the Odin satellite: An overview, Planetary and Space Science, 55, 1093 – 1102, https://doi.org/ 10.1016/j.pss.2006.11.021, 2007.

von Clarmann, T., Höpfner, M., Kellmann, S., Linden, A., Chauhan, S., Funke, B., Grabowski, U., Glatthor, N., Kiefer, M., Schieferdecker, T., Stiller, G. P., and Versick, S.: Retrieval of temperature, $H_2O$, $O_3$, $HNO_3$, $CH_4$, $N_2O$, $ClONO_2$ and ClO from MIPAS reduced resolution nominal mode limb emission measurements, Atmospheric Measurement Techniques, 2, 159 – 175, 2009.

Weigel, K., Rozanov, A., Azam, F., Bramstedt, K., Damadeo, R., Eichmann, K.-U., Gebhardt, C., Hurst, D., Kraemer, M., Lossow, S., Read, W., Spelten, N., Stiller, G. P., Walker, K. A., Weber, M., Bovensmann, H., and Burrows, J. P.: UTLS water vapour from SCIAMACHY limb measurements V3.01 (2002-2012), Atmospheric Measurement Techniques, 9, 133 – 158, https://doi.org/10.5194/ amt-9-133-2016, 2016.

---

## Author Comment (AC2) · 17 Feb 2019

**Response to the Comments**
* * *
Color code:

comments of the reviewer
response by the authors
proposed changes in the manuscript
* * *
General comment:

The paper has an impressive author list and I am sure that the results are worthwhile to publish. But in my mind the pdf-format is not a best way to publish figures including an impenetrable jungle of line plots. You would need a shorter summary in pdf and then an interactive web-page where you could extract detailed results.

General response:

Certainly interactive figures would be a better way to present the data. There are even technologies to include interactive features in PDF files, e.g. U3D (see https://en.wikipedia.org/wiki/Universal_3D). However, since Copernicus does not support such interactive PDF files this is no solution. To set up a website with interactive figures can be achieved with reasonable effort. But, also such a website cannot act as an official part of a Copernicus publication. It can only be a supplement, not a replacement.

A shorter summary is very difficult to achieve. We attempt to do justice to 33 data sets, resulting not only literally in a million figures. How to condense all this information into a sustainable amount has been part of many and quite sustained discussions within the WAVAS-II core team. We are always open to new ideas. Since the critique here is rather vague we opt to work with the comments that the other reviewer provided on this topic.

Shortening of Sect. 5.4.

---

## Author Response (AR2)

Dear Helen,

please find our revised version of the manuscript, which considers the comments of reviewer #1.

With kind regards

Stefan on behalf of all authors

**Response to the Comments**
* * *
Color code:

comments of the reviewer

response by the authors

proposed changes in the manuscript
* * *
Comment #1:

The non-linearity issue already mentioned in my 1st review. I like the text you have provided in 3.3 and in particular the warning that there is non-intuitive behaviour. BUT: It is quite conceivable that the implication of this non-intuitive behaviour is overlooked. On page 16, now line 29, I suggest not only to write ... or $\pm 100\%$ on some occasions, but to explain here as an example of the non-intuitivity: this means that $x_1$ and $x_2$ differ by more than their arithmetic mean value. To my opinion this would clarify the issue.

Response #1:

We have decided to move this example to an appendix. A reference to the appendix is provided in the description of the relative biases and the associated non-intuitive behaviour.

Creation of an appendix.

Comment #2:

I am also please by the list of bullet points in the summary section which helps the user to find what s/he needs. BUT: it can be improved. I give some examples:

a) In the paper you use one-to-one comparisons and the meaning of bias is clear in this case. Now we are interested in the overall picture. So what does it mean then if we read that, for instance, ACE-FTS has a -10% bias? Relative to what? Perhaps one can find it by scrolling back and forth in the paper, but it would be better to have the important information at this place.

b) Also expressions like "show slightly more pronounced negative biases..." should be avoided. "Slightly more" than what? In this case the "slightly more" can just be deleted. But there are similar phrases. Please correct them all.

c) Page 29, lines 18/19: It is nice that the other three datasets are mostly positive, but probably you mean something different. Also the next sentence is a bit strange.

Response #2:

a) How the relative biases were calculated was added to the beginning of the Conclusion section.

The corresponding text has been extended.

b) As suggested, the phrase "slightly" has been avoided. This concerns the bias summary for the ACE-FTS and SOFIE data sets.

The phrases have been removed.

c) It was, of course, meant that these three SCIAMACHY data sets primarily exhibit positive biases.

The text has been adapted accordingly.

[revised manuscript text omitted]